# Cell reprogramming shapes the mitochondrial DNA landscape

Wei Wei[1,2], Daniel J. Gaffney ⬡ [3,4] & Patrick F. Chinnery ⬡ [1,2✉]

Individual induced pluripotent stem cells (iPSCs) show considerable phenotypic hetero-geneity, but the reasons for this are not fully understood. Comprehensively analysing the mitochondrial genome (mtDNA) in 146 iPSC and fibroblast lines from 151 donors, we show that most age-related fibroblast mtDNA mutations are lost during reprogramming. However, iPSC-specific mutations are seen in 76.6% (108/141) of iPSC lines at a mutation rate of 8.62 $\times 10^{-5}$/base pair. The mutations observed in iPSC lines affect a higher proportion of mtDNA molecules, favouring non-synonymous protein-coding and tRNA variants, including known disease-causing mutations. Analysing 11,538 single cells shows stable heteroplasmy in sub-clones derived from the original donor during differentiation, with mtDNA variants influencing the expression of key genes involved in mitochondrial metabolism and epidermal cell dif-ferentiation. Thus, the dynamic mtDNA landscape contributes to the heterogeneity of human iPSCs and should be considered when using reprogrammed cells experimentally or as a therapy.

[1] Department of Clinical Neuroscience, University of Cambridge, Cambridge, UK. [2] Medical Research Council Mitochondrial Biology Unit, School of Clinical Medicine, University of Cambridge, Cambridge, UK. [3] Human Induced Pluripotent Stem Cell Initiative, Wellcome Genome Campus, Hinxton, UK. [4] Wellcome Sanger Institute, Wellcome Genome Campus, Hinxton, UK. ✉email: pfc25@cam.ac.uk

There is a growing interest in the use of induced pluripotent stem cells (iPSCs) to model human disease mechanisms and in cell-based therapies, but iPSCs derived from the same tissue show considerable phenotypic heterogeneity that can affect their capacity to differentiate into organ-specific lineages. Bulk and single-cell analysis has shown common genetic variation contributes to differences in gene expression profiles between donors[1–3], but the majority of the variation remains unexplained. Even iPSCs derived from the same donor cell line show phenotypic differences that are poorly understood. However, independently differentiating the same cell line several times reproduces similar cell phenotypes[4], implicating cell-intrinsic factors driving the heterogeneity, rather than the external environment.

Although there has been considerable effort describing nuclear genetic and epigenetic differences that arise during reprogramming, the 16.5 Kb mitochondrial genome (mtDNA) remains largely uncharacterised. mtDNA codes for 13 essential peptide components of the oxidative phosphorylation systems, and 2 rRNAs required for intra-mitochondrial protein synthesis[5]. Most human cells contain >1000 copies of mtDNA, and deep sequencing has shown that most humans harbour a mixed population of mitochondrial genomes (heteroplasmy)[6]. mtDNA mutations can arise de novo during life or be inherited down the maternal line, and specific mtDNA variants compromise oxidative metabolism and the synthesis of adenosine triphosphate (ATP). High percentage levels of specific mtDNA mutations cause severe multi-system metabolic diseases that affect ~1 in 8000 humans, but they also contribute to the pathology of common age-related disorders, including Parkinson's disease[5]. mtDNA mutations influence canonical cell signalling pathways, reactive oxygen species production, amino acid metabolism and cell growth through indirect effects on nuclear gene transcription[7–10].

mtDNA mutations also accumulate in somatic tissues throughout life, raising the possibility that mtDNA variation contributes to the molecular heterogeneity of human iPSCs. In keeping with this, several small studies have described mtDNA variants present in iPSC lines that were not detected in parental fibroblast lines, with some having detrimental effects on metabolism[11], including mitochondrial respiration in derived cardiomyocytes[12]. However, with <25 donors studied to date, the landscape of mtDNA variation in human iPSCs poorly understood[13].

In this work, we analysed high-depth mtDNA sequences in 146 iPSC lines as part of the Human-Induced Pluripotent Stem Cells Initiative (HipSci). We show extreme mtDNA diversity, with selection favouring some variants during reprogramming. Single-cell analysis shows that mtDNA variants define sub-clones that modulate gene expression within iPSCs and subsequent differentiated cell lineages.

## Results

**Data collection and quality control**. Initially, we analysed high-depth mtDNA sequences derived from whole-genome sequence (WGS) data on 146 iPSC and 151 fibroblast lines obtained from 151 different donors aged 27–77 years recalled through the NIHR BioResource as part of the Human Induced Pluripotent Stem Cells Initiative (HipSci, http://www.hipsci.org/). After sample quality control (QC), 141 iPSC lines and 146 fibroblast lines were included in the analysis (see 'Methods'), with one iPSC line from 25 donors and 2 iPSC lines from 58 donors included (Fig. 1a, Supplementary Fig. 1a and Supplementary Data 1). The average WGS depth was 44-fold (s.d. = 9-fold, and mtDNA depth was 1824-fold (s.d. = 2249-fold) (Supplementary Fig. 2a, i, j). There was no detectable difference in the depth of mtDNA sequencing

between the fibroblasts and their derived iPSCs (median depth in fibroblasts 777×, median depth in iPSCs 887×, $P = 0.907$, paired Wilcoxon test) (Supplementary Fig. 2j), consistent with similar numbers of cells sequenced between each fibroblast and iPSC pair. First, we identified high-quality mtDNA variants relative to the revised Cambridge Reference Sequence (rCRS)[14], including variants in mixed proportions in the bulk sequencing (heteroplasmic variants). We filtered out calls likely to be due to systematic errors, including nuclear-encoded mitochondrial DNA sequences (NUMTs) and length variants flanking polynucleotide tracts (see 'Methods'). We then determined the frequency distribution of major population-specific mtDNA haplogroups (macro-haplogroups) (Supplementary Fig. 1b), and compared this to published data[15] ($P < 2.2 \times 10^{-16}$, $R^2 = 0.99$, Pearson's correlation test), confirming that the vast majority of donors were representative of the United Kingdom population. In total, 143 of the 146 donors (98%) belonged to one of the macro-haplogroups H, I, J, K, T, U, V, or W. As a further QC step, we only included heteroplasmic variants with bulk heteroplasmy allele fractions (HFs) between 2 and 98% (see 'Methods').

Heteroplasmy calling was validated using bulk RNA-sequencing (RNA-seq) data generated from 102 iPSCs (see 'Methods'). As expected, the depth of RNA-seq was less uniform than WGS (Supplementary Fig. 2b). However, the average depth of mtDNA was deep sufficient (4508-fold, s.d. = 2946-fold) to detect low-level bulk heteroplasmic variants. The HFs of heteroplasmic variants detected by both techniques were strongly correlated ($P = 6.75 \times 10^{-120}$, $R^2 = 0.94$, Pearson's correlation test) (Supplementary Fig. 2d).

**Distinct spectrum of mtDNA mutations in fibroblasts with age**. In total, 143 (98%) of 146 fibroblast lines carried at least one heteroplasmy on bulk analysis (HF $> = 2\%$). The mean number of heteroplasmic mtDNA variants per fibroblast line was 8 (s.d. = 4), with a greater mean number in fibroblasts from older individuals ($N = 146$, $P = 4.92 \times 10^{-3}$, coefficient estimate = 0.10, s.d. = 0.034, linear regression model) (Figs. 1b and 2a, b), where the correlation was driven in part by variants in the D-loop region ($P = 9.41 \times 10^{-5}$, coefficient estimate = 0.04, s.d. = 0.010, linear regression model) (Fig. 2c). Overall, the mean HF per fibroblast line was 9.10% (s.d. = 7.40%). The mean HF per fibroblast line of the D-loop variants increased with the age of the donor ($P = 8.49 \times 10^{-3}$, coefficient estimate = 0.025, s.d. = 0.009, linear regression model), but there was a neither detectable correlation with age seen for the other genomic regions nor across the whole mtDNA genome ($P > 0.05$, linear regression model) (Fig. 2d, e, f).

Eight heteroplasmic variants were present in >5% of the fibroblast lines (Figs. 1b and 2g, h). m.414G was the most common recurrent mutation (28 fibroblasts lines, 19.2%), with the proportion of individuals carrying this variant increasing with age (Fig. 2i) as noted before[16]. The previously observed m.2623G and m.13369C variants[13] were also seen in 12.3%, despite being exceptionally rare in blood from the same population[6]. These mutations occurred on different mtDNA haplogroup backgrounds, indicating recurrent mutation events.

Each mutation initially affects a single mtDNA molecule, but to be detected by our WGS pipeline, at least 2% of molecules must be affected in the bulk DNA sample (where 2% is the HF threshold we used to detect heteroplasmic variants, see 'Methods'). Thus, the mutations we observed are a consequence of the original mutation event followed by subsequent copying (clonal expansion) and propagation of the variant into many daughter cells. Although it is conceivable that some high-level

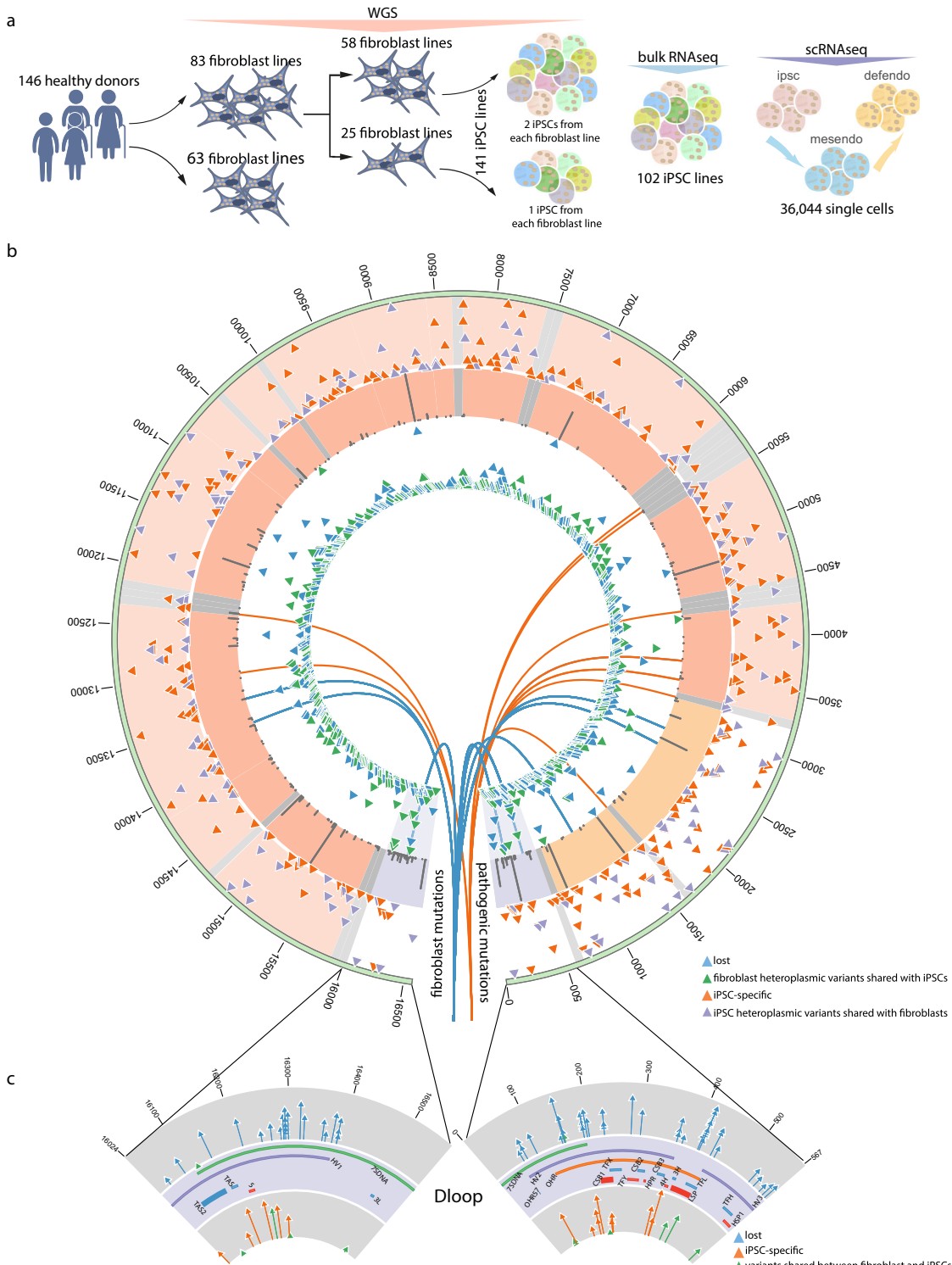

**Fig. 1 mtDNA heteroplasmic variants detected from whole-genome sequences. a** Summary of bulk whole-genome (WGS), bulk RNA and single-cell RNA (scRNA) sequencing data analysed in this study. Colour represents the heterogeneity of cell populations under WGS and bulk RNA-seq, and three different cell stages under scRNAseq, where mesendo = mesoderm, and defendo = definitive endoderm. **b** Heteroplasmic variants detected from WGS. Circos plot from outside the circle to inside: (1) mtDNA position; (2) heteroplasmic variants identified in 141 iPSC lines; (3) minor allele frequency of common variants (MAF > 1%) in European population[55]; mtDNA genes (purple— D-loop, red—coding region, yellow—rRNAs and grey—tRNAs); (4) heteroplasmic variants identified in fibroblast cell lines; (5) blue lines pointing to the positions of variants specific in fibroblast cell lines; (6) orange lines pointing to the positions of pathogenic mutations; (2) & (4) vertical axes represent the HFs; (3) vertical axes represent the frequencies of single-nucleotide substitutions. **c** High resolution of mtDNA D-loop region. Plots from top to bottom: (1) mtDNA position; (2) decrease shift variants; (3) D-loop regions; (4) increase shift variants; (2) & (4) vertical axes represent heteroplasmic shifts.

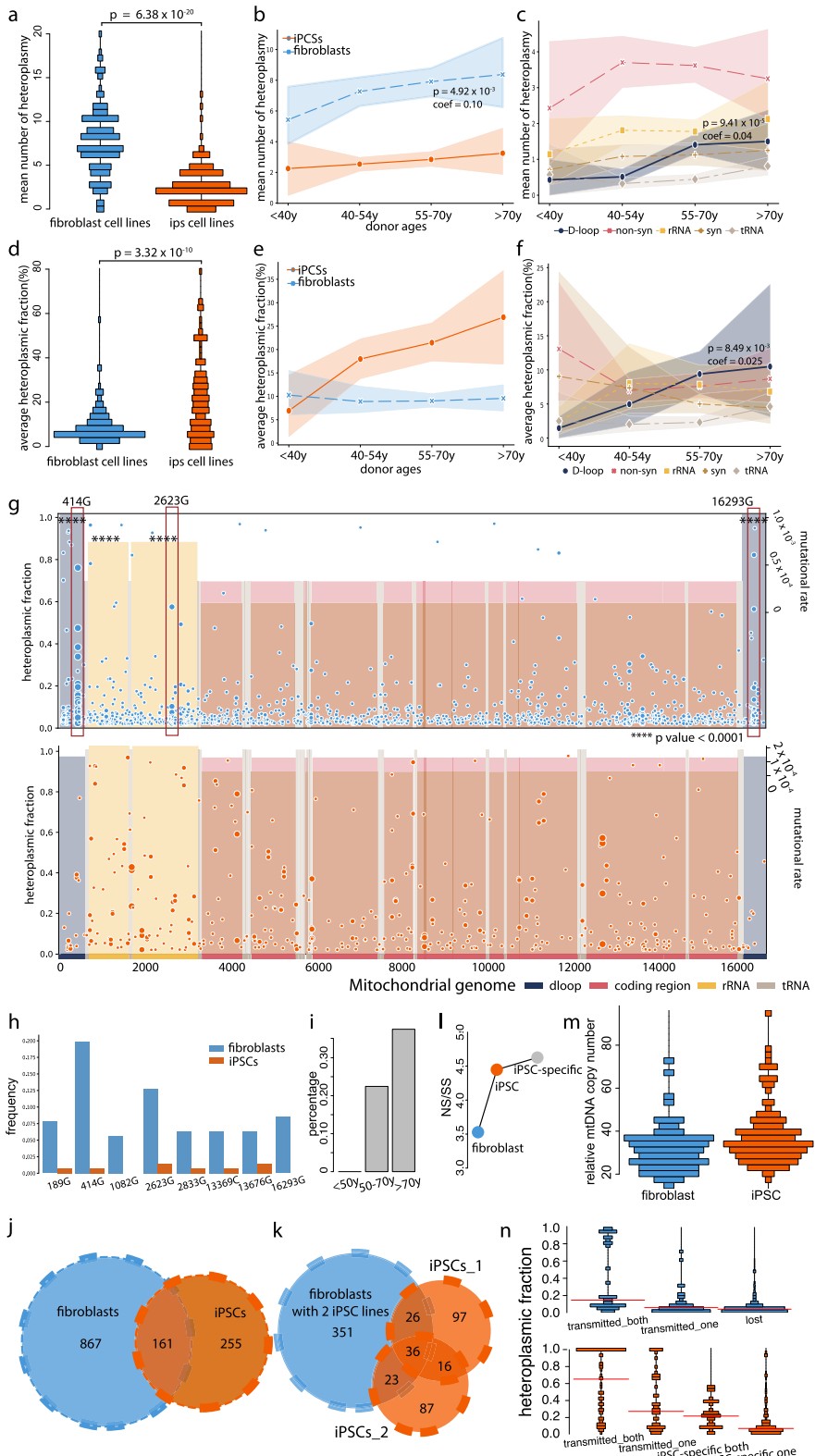

heteroplasmic variants could reflect recurrent mutations in the same individual, this seems unlikely because the majority (64.5%) of heteroplasmic variants were seen in only one fibroblast line. Clonal expansion must therefore be playing a key role in generating the higher heteroplasmy values seen in fibroblasts from older subjects, and explain the correlation between the HF and age for m.414G (Fig. 2i).

**Cell reprogramming changes the spectrum of mtDNA mutations**. Next, we carried out a similar analysis of high-depth mtDNA sequences in 141 iPSCs derived from 83 of the original fibroblast lines (Fig. 1a, Supplementary Fig. 1a and Supplementary Data 1). In all, 130 (92%) of 141 iPSCs carried at least one heteroplasmy (HF >= 2%). The mean number of heteroplasmic variants per iPSC line was 3 (s.d. = 2), with a mean HF of 20.7%

**Fig. 2 Characteristics of mtDNA heteroplasmic variants detected through the bulk analysis of 146 fibroblast cell lines and 141 derived iPSC lines.**
**a** Distribution of the mean number of heteroplasmies defined in fibroblast and iPS cell lines. **b** Correlation of the mean number of heteroplasmies per fibroblast cell line with the donor's age. Shaded regions show mean ± standard deviation (s.d.). **c** Correlation of the mean number of heteroplasmies per fibroblast cell line in each mtDNA region with the donor's age. Shaded regions show mean ± s.d. **d** Distribution of the average heteroplasmy fraction (HF) in fibroblast and iPS cell lines. **e** Correlation of the average HF per fibroblast cell line with the donor's age. Shaded regions show mean ± s.d. **f** Correlation of the average HF per fibroblast cell line in each mtDNA region with the donor's age. Shaded regions show mean ± s.d. **g** Heteroplasmies defined in fibroblast (top) and iPS cell lines (bottom). HFs are shown on the left y axis. mtDNA regions covered by different colours. The depth of the shading represents the mutation rate of each mtDNA region (shown on the right side of y axis). Three fibroblast-specific mutations are highlighted in red rectangles. The regions were significantly enriched mutations than expected by chance were labelled by asterisks. **h** Frequency of specific mutations in fibroblast and iPS cell lines. **i** The fibroblast-specific mutation 414G was associated with the donor's age. **j** Distribution of heteroplasmies defined in fibroblast and iPS cell lines. **k** Distribution of heteroplasmies defined in fibroblast and iPS cell lines, two iPSC lines derived from the same fibroblast cell line are shown separately. **l** Ratio of non-synonymous/synonymous variants (NS/SS) observed in fibroblasts, iPSCs and iPSC-specific variants. **m** Distribution of the mtDNA copy number in fibroblast and iPS cell lines. **n** Distribution of the heteroplasmy fraction in fibroblast and iPS cell lines. Red lines show the mean HFs within each dataset. **a**, **d** P values were calculated using two-sided Wilcoxon test. Source data are provided as a Source Data file. **b**, **c**, **f** P values were calculated using linear regression model. **g** P values were calculated using two-sided Fisher's exact test.

(s.d. = 17.4%) (Figs. 1b and 2a, d, g). In contrast to the skin fibroblasts, there was no detectable correlation between the age of the individual at the time of the skin biopsy and the mean number of heteroplasmic variants. Although there was a trend between donors' age and average HF per person across the whole mtDNA genome, our observations did not provide statistical support for this trend (Fig. 2e and Supplementary Fig. 3a).

Overall, the iPSC lines contained less heteroplasmic variants than their fibroblast parental lines ($P = 6.38 \times 10^{-20}$, paired Wilcoxon test) (Fig. 2a and Supplementary Fig. 3b). In total, 15.7% (161/1028) of the heteroplasmic variants observed in the fibroblasts were 'shared' with at least one of their derived iPSC lines, and 84.3% (867/1028) of the variants present were not detected at HFs >= 2% in the iPSCs ('lost' mutations) ($P < 2.2 \times 10^{-16}$, 95% CI 0.213–0.262, exact binomial test) (Fig. 2j, k). By contrast, 62.6% (270/431) of the variants present in 76.6% (108/141) of iPSC lines were not detected at HF >= 2% in the donor fibroblasts (classified here as 'iPSC-specific' mutations) (Fig. 2j), making a greater contribution to the mtDNA landscape in iPSCs than shared variants ($P = 4.69 \times 10^{-6}$, 95% CI = 0.564–0.660, exact binomial test). There was no detectable difference in either the mean HF, the mean number of heteroplasmic variants, or the mean number of iPSC-specific mutations in iPSC lines analysed at different passages ($P = 1$, pairwise Wilcoxon test) (Supplementary Figs. 1c and 4).

Next, we compared the mutation frequency across different mtDNA regions. In fibroblasts, the D-loop had a higher mutation frequency than expected by chance, averaging across the whole mtDNA (mean = $1.04 \times 10^{-3}$/base pair, D-loop vs. expected $P = 2.03 \times 10^{-16}$, 95% CI 1.84–2.63, Fisher's exact test). The D-loop had a higher mutation frequency than all of the other mtDNA regions (D-loop vs. rest of regions $P < 2.2 \times 10^{-16}$, 95% CI 1.95–2.71, Fisher's exact test) (Fig. 2g). Heteroplasmic variants were also enriched in the two rRNA genes (mean = $7.45 \times 10^{-4}$/base pair, rRNA vs. expected $P = 8.54 \times 10^{-10}$, 95% CI 1.37–1.83, Fisher's exact test). By contrast, in the iPSCs, the mutation frequency in the D-loop was lower than the rest of the mtDNA ($P = 3.55 \times 10^{-6}$, 95% CI 1.71–4.54, Fisher's exact test) (Fig. 2g), and overall, there was a higher mutation frequency for non-synonymous variants ($P = 1.11 \times 10^{-3}$, 95% CI 0.54–0.86, Fisher's exact test) (Fig. 2g). In keeping with this, the ratio of non-synonymous (NS) to synonymous (SS) coding region variants was greater for the heteroplasmic variants in iPSCs than the heteroplasmic variants in fibroblasts (iPSCs: 4.45, fibroblast: 3.52), and significantly higher than previously reported in blood[6] (NS/SS in blood 1.30, fibroblast vs blood $P < 2.2 \times 10^{-16}$, iPSC vs blood $P = 2.28 \times 10^{-14}$, Fisher's exact test) (Fig. 2l). Taken together, these findings indicate that cell reprogramming shaped

the spectrum of mtDNA heteroplasmic variants seen in the fibroblast lines. The change in profile was largely driven by the loss of variants from the fibroblasts and iPSC-specific variants appearing in the iPSC lines (Fig. 3a) with the greatest NS/SS ratio (Fig. 2l).

**Mechanistic insights from two iPSCs from the same donor.** To gain further insight into the mechanisms, we looked at the 116 iPSCs where two lines had been derived from 58 fibroblast donors. Only 8.3% (36/436) of the unique heteroplasmic variants present in each donor fibroblast were detected in both iPSC lines (shared both); 11.0% (49/436) were present in one of the two iPSC lines (shared one/lost one); and 80.6% (351/436) were not detected in either iPSC line (lost from both). By contrast, 78.0% (184/236) of the unique heteroplasmic variants in each donor iPSC line were only present in one of the two iPSCs (iPSC-specific one) and 6.8% (16/236) in both iPSCs (iPSC-specific both) (Fig. 2k). Thus, the vast majority of mutations present in the fibroblast lines were lost on reprogramming, and the iPSC-specific mutations most likely occurred independently each time a fibroblast line was reprogrammed. To determine whether some of the iPSC-specific mutations were actually present in the fibroblasts but below our detection threshold (HF < 2%), we re-analysed the fibroblast data without a lower cut-off filter for HFs. This showed that a minority (14.8%; 40/270) of the previously classified iPSC-specific variants were actually present in the original fibroblast line at HF < 2%. Factoring this into account, the iPSC-specific mutation rate was $8.62 \times 10^{-5}$ per base pair per genome per reprogramming ($1.87 \times 10^{-6}$ per base pair per mtDNA molecule normalising for the average number of mtDNA molecules per cell), which is a conservative estimate based on mutations with a HF >= 2% in the iPSCs and a HF > 0 in their matched fibroblast cell lines. Finally, to determine whether our initial detection threshold had a major impact on the measured iPSC-specific mutation rate, we decreased the detection threshold in iPSCs by 0.5% increments. This showed the anticipated increased mutation rate, but only by approximately fourfold, indicating that our approach is reliable to within one order of magnitude (Supplementary Fig. 3c).

The rapid segregation of mtDNA variants we observed between fibroblasts and iPSCs is consistent with a bottleneck during reprogramming. This resembles observations during germ cell development, where a reduction in cellular mtDNA content contributes to the rapid segregation of heteroplasmic mtDNA variants[17,18]. A similar genetic bottleneck effect could explain why variants not detectable in fibroblasts become detectable in iPSCs. However, given the inherent inefficiency of cellular reprogramming, an alternative explanation is that there is a

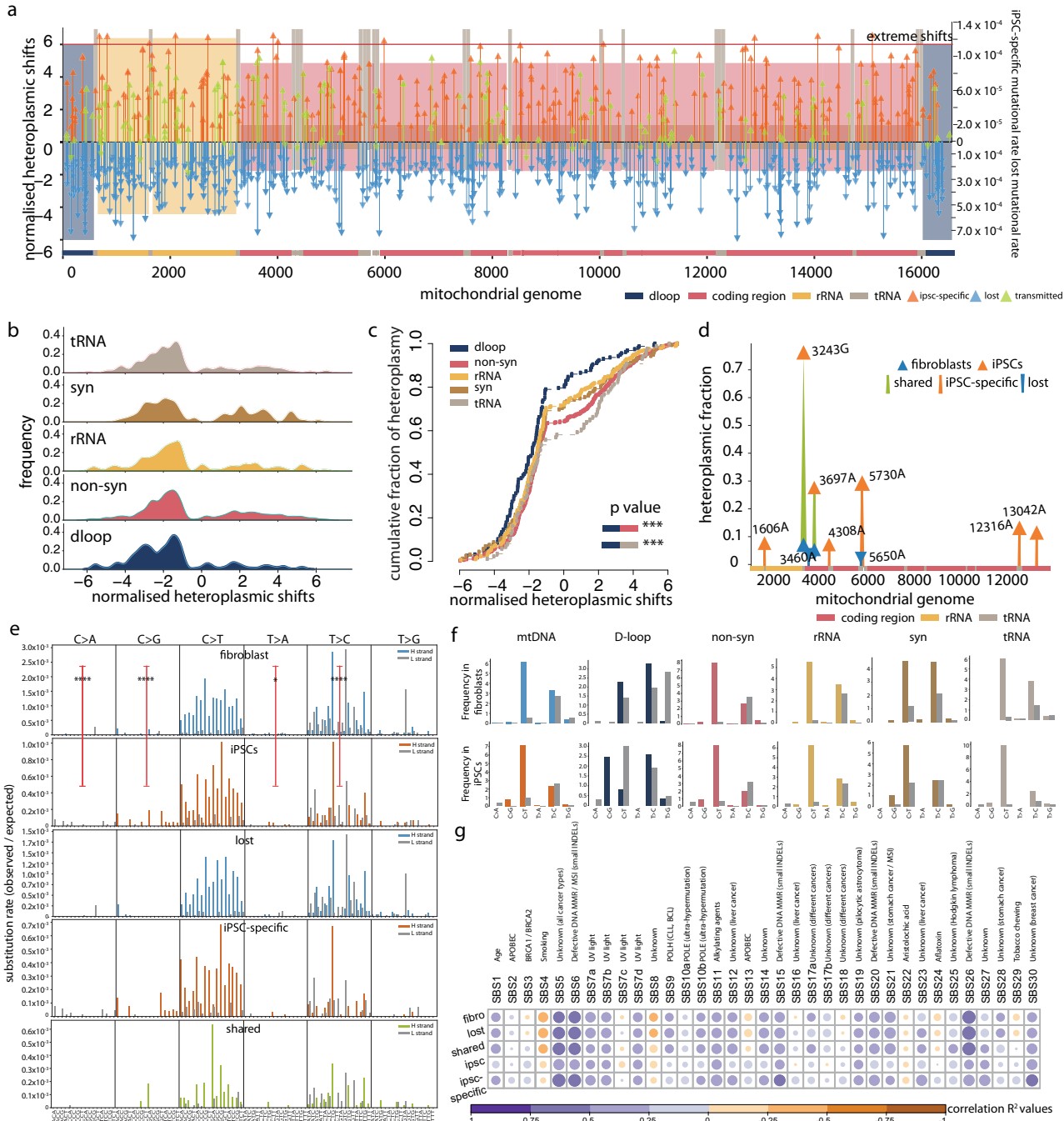

**Fig. 3 The spectrum of mtDNA mutations changes during cell reprogramming. a** Normalised heteroplasmic shifts (HS) estimated between 83 fibroblast cell lines and 141 derived iPSC lines. Normalised heteroplasmic shifts are shown on the left side of the y axis. mtDNA regions shown by different colours; the depth of shading represents the mutation rate of iPSC-specific variants and lost variants within each mtDNA region (shown on the right side of the y axis). Variants above the red lines were extreme shifts seen in iPSCs. Source data are provided as a Source Data file. **b** Distributions of the HS in different mtDNA regions. **c** Cumulative distributions of the HSs within each mtDNA region. P values were calculated using two-sided Fisher's exact test. **d** Pathogenic mutations observed in this study. Different colours represent lost, iPSC-specific or shared mutations between fibroblast cell lines and derived iPSC lines. Blue triangles are HFs in fibroblasts and orange triangles are HFs in iPSCs. mtDNA regions shown at the bottom. **e** Trinucleotide mutational signature of heteroplasmic variants observed in fibroblasts, iPSCs, lost in fibroblasts, iPSC-specific and shared variants between fibroblasts and iPSCs. The bars represent the substitution rate, mutations from the H or L-strand are shown in different colours. P values were calculated using two-sided Fisher's exact test. **f** Mutational signature of six categories (C > A, C > G, C > T, T > A, T > C & T > G) within each mtDNA region. The bars represent the relative frequency of each category. Mutations from the H or L-strand are shown in different colours. **g** Correlation between mutational signatures observed in this study with the cancer signatures. The gradients of circles correspond to correlation $R^2$ values. The sizes of circles correspond to the p values (larger circles have smaller P values). The names of cancer signatures are shown at the top.

'cellular bottleneck', where only a small proportion of the fibroblasts contribute to the iPSC cell population. In keeping with this, the average amount of mtDNA per cell was greater in iPSCs than in the matched fibroblast lines ($P = 0.08$, paired Wilcoxon test) (Fig. 2m), although we cannot exclude the possibility of a reduction in cellular mtDNA content at precisely the time of reprogramming. Of note, the cell lines were not physically sub-cloned during the iPSC generation.

**Changes in heteroplasmy fraction (HF) during reprogramming.** As expected, the HF for shared heteroplasmic variants found in both the fibroblasts and iPSCs lines was greater than HF seen for lost and iPSC-specific heteroplasmic variants (shared vs lost in fibroblasts $P = 4.75 \times 10^{-14}$, shared vs iPSC-specific in iPSCs $P < 2.2 \times 10^{-16}$, Wilcoxon rank-sum test) (Fig. 2n), in keeping with a genetic bottleneck effect. Next, we studied the change in HF for each variant during reprogramming, normalising each shift to the original HF in the founder fibroblast line (Fig. 3a) (see 'Methods'). The normalisation is important because the simple difference in HF between the fibroblast and iPSC lines does not reflect the magnitude of the fold change in heteroplasmy (e.g., doubling the HF from 2–4% would give the same absolute change as 50–52%). In addition, the difference between the original fibroblast line and the iPSC is limited by 0 and 100%, which is particularly important for low-level heteroplasmic variants[6]. Subsequent analysis was therefore performed on log2 ratio of HF between the iPSC line and the original fibroblast line, which we termed the 'heteroplasmic shift' (HS). Note that this correction reflects the fold change in HF, but the size of this change is influenced by the initial HF.

Despite the overall loss of heteroplasmic variants (Figs. 2j and 3a), the mean HF was significantly higher in iPSC lines than the fibroblast lines (individual fibroblast-iPSC pairs $P = 3.32 \times 10^{-10}$, paired Wilcoxon test) (Fig. 2d). Of 161 shared heteroplasmic variants, 138 increased and only 23 decreased on reprogramming ($P < 2.2 \times 10^{-16}$, 95% CI = 0.793–0.907, exact binomial test) (Fig. 3a). In total, 12 variants observed in 9 iPSC lines showed extreme increase shifts (normalised shift values >6) lying outside the overall distribution. Five of these 12 variants were non-synonymous variants, and none was in the D-loop (Fig. 3a). Consistent with the previously observed loss of mtDNA heteroplasmic variants, in all mtDNA regions, HS were more likely to be negative than positive on reprogramming (Fig. 3b and Supplementary Data 2). Next, we compared the HS between different mtDNA regions. The HS for D-loop was more likely to decrease than other regions (2.24-fold decrease, $P = 9.48 \times 10^{-5}$, Fisher's exact test), and the HSs for non-synonymous and tRNA variants were more likely to increase than other regions (non-synonymous 1.43-fold increase, $P = 3.90 \times 10^{-3}$; tRNA 1.61-fold increase, $P = 4.12 \times 10^{-2}$, Fisher's exact test) (Fig. 3c and Supplementary Data 2).

To explore the possibility that confounding variables explained these trends, we modelled the direction of HS (increase shift/ decrease shift) of a variant as a function of the age of the individual when the skin biopsy was taken, the HF in the original fibroblast line, the mitochondrial genome location, and the macro-haplogroup of the donor (see 'Methods'). Variants in the D-loop region were less likely to increase than variants in non-synonymous and tRNA regions (non-synonymous $P = 1.15 \times 10^{-3}$, coefficient estimate = 0.78, s.d. = 0.24, tRNA $P = 4.93 \times 10^{-3}$, coefficient estimate = 0.89, s.d. = 0.32, logistic regression model) (Fig. 3a, c). Thus, mtDNA heteroplasmy fraction is modified during reprogramming, with heteroplasmy tending to increase or decrease in specific regions of the mtDNA, but with an overall increase in the level of heteroplasmy in iPSCs.

The preferential loss of D-loop variants and propagation of non-synonymous and tRNA variants are the complete opposite of what is seen during germline transmission[6]. Thus, tissue or context-specific factors influence the segregation of mtDNA heteroplasmy in humans, implying a more complex mechanism than simply the effect of the variants on oxidative phosphorylation and ATP synthesis. This is in keeping with work in mice, where pluripotent cells modulate heteroplasmy in a different way to mouse embryonic fibroblasts (MEFs), and reprogramming MEFs to iPSCs recapitulates the pluripotent cell behaviour[19]. For the D-loop, one potential explanation involves the selection against variants that compromise mtDNA replication, given the increased mtDNA content seen in iPSC lines. To gain insight into the selection of the D-loop region, we further defined the heteroplasmic variants in different D-loop regions (Fig. 1c). Variants in the origin of heavy-strand replication (*MT-OHR*) were more likely to be lost than expected when compared to the average across D-loop region ($P = 8.76 \times 10^{-5}$, Fisher's exact test), particularly in L-strand promoter (*MT-LSP*) ($P = 7.26 \times 10^{-7}$, Fisher's exact test) which generates the RNA–primer required for DNA synthesis and includes the age-related mutation m.414G. By contrast, iPSC-specific mutations were enriched in *MT-OHR* ($P = 7.59 \times 10^{-3}$, Fisher's exact test), particularly in *MT-CSB1* ($P = 6.28 \times 10^{-3}$, Fisher's exact test). Although it is difficult to explain all of these patterns at present, their distribution is likely to reflect the function of D-loop structures.

**Potential pathogenic mutations.** Of the 12 known pathogenic mtDNA mutations present in 9 donor fibroblast lines (Figs. 1b and 3d), five were lost during reprogramming, including two fibroblast lines where the mutations (m3460G> A and m.3697G> A) were not present in 2 iPSC lines derived from each donor. By contrast, m.3243A > G and m.3697G > A were found in two separate fibroblast lines, shared by both of the iPSC lines derived from each donor, and in each case with higher HF.

Five iPSC-specific pathogenic mutations were not seen in the fibroblast lines from four donors, giving a iPSC-specific pathogenic mutation rate of $2.14 \times 10^{-6}$ per base pair per genome (95% CI $7.88 \times 10^{-7}$–$5.31 \times 10^{-6}$ per base per genome) ($6.82 \times 10^{-8}$ per base pair per mtDNA molecule normalising for the average number of mtDNA molecules per cell). Accounting for differences in the definition of a pathogenic mutation, the iPSC-specific mutation rate during iPSC reprogramming was ~5.9-fold greater than estimated for the germ line[6].

**Reprogramming shapes the mtDNA mutational signature.** Building on previous observations in cancer and the germ line[6,20,21], next we determined the mutational signature of the heteroplasmic mtDNA variants. As expected, in fibroblasts, C > T and T > C substitutions were the most common. We did not observe the L-strand T > C mutation bias seen in whole blood and 18 cancer types[20] (Fig. 3e, f), however, the same pattern has been noted before in melanoma[22] and the germline variants[23]. Unlike the nuclear genome[24], mtDNA did not show the signature of ultraviolet light-induced DNA damage, implying an alternative mechanism of mutagenesis and/or additional factors shaping the trinucleotide signature over time. By contrast, the iPSCs had a distinct trinucleotide mutational signature (iPSCs vs fibroblasts $P = 3.66 \times 10^{-42}$, Stouffer's method for combining Fisher $P$ values), with more C > A, C > G, T > A variants ($P = 9.47 \times 10^{-7}$, 95% CI 0.036–0.304, $P = 3.00 \times 10^{-30}$, 95% CI 0.020–0.088, $P = 0.03$, 95% CI 0.193–0.968, Fisher exact test) and less T > C variants ($P = 4.07 \times 10^{-26}$, 95% CI 2.81–4.73, Fisher exact test) than donor fibroblasts (Fig. 3e). Surprisingly, the C > A

substitutions were preferentially seen on the L-strand in iPSCs (observed vs. expected $P = 5.36 \times 10^{-3}$, 95% CI 1.725–Inf, Fisher exact test), and C > G substitutions were strongly enriched on the H-strand (observed vs. expected $P = 1.35 \times 10^{-7}$, 95% CI $6.68 \times 10^{-4}$–0.18, Fisher exact test) (Fig. 3e). The L-strand C > T substitution bias was prominent in the D-loop region, which showed an opposite trend to fibroblasts and cancer somatic mutations (Fig. 3f and Supplementary Fig. 5a)[20]. iPSC-specific heteroplasmic variants made the greatest contribution to the distinct iPSC mutational signature (iPSC-specific vs lost: $P = 1.18 \times 10^{-12}$; iPSC-specific vs shared: $P = 0.01$, Stouffer's method for combining Fisher $P$ values), with the shared and lost variants having the similar signature profile ($P = 0.06$, Stouffer's method for combining Fisher $P$ values) (Fig. 3e and Supplementary Fig. 5b). Thus, cell reprogramming shapes the overall mutational signature in iPSCs. The pattern of iPSC-specific mutations resembles the signature seen when there is disordered mismatch repair (MMR)[24] (Fig. 3g), although the absence of MMR within mitochondria implicates alternative mechanisms.

**mtDNA variants detected using single-cell RNA sequencing**. We detected the mtDNA variants in 36,044 single-cell transcriptomes (scRNAseq) from 125 donors at three canonical stages of endoderm differentiation: iPSCs ($n = 9661$), mesoderm (mesendo, $n = 10,199$), definitive endoderm (defendo, $n = 9906$) and undefined cells ($n = 6278$)[3] (see 'Methods'). We studied 11,538 single cells (iPSCs 2711, mesendo 3402, defendo 3796 and undefined 1629) from 60 donors. The bulk WGS data of 60 progenitor fibroblast cells were also available and included in our analysis above. In total, 6288 of 11,538 single cells from 36 iPSC lines also had the bulk WGS data from the same iPSC lines. In all, 5607 cells from 30 iPSC lines had matched bulk RNA-sequencing data (Fig. 1a, Supplementary Fig. 1d and Supplementary Data 1), allowing several cross-validation steps.

The average depth of each mtDNA gene and rRNA was 66 to 1144-fold (Fig. 4a and Supplementary Fig. 2c). After excluding regions with sequencing depth below 200x, we detected 173,136 variants (HF > 2%) in 11,538 single cells across 11,704 bp (2463 bp of rRNAs and 9241 bp of coding regions) (Fig. 4a, b shows the data of 9909 cells from the three cell stages). In total, 139,778 variants were homoplasmic (HF > 95%). All of the observed homoplasmic variants were also detected in their matched fibroblast cells by WGS. In all, 1989 of the 139,778 homoplasmic variants (1.4%) were heteroplasmic in their matched fibroblast cells, with the HF increasing to >95% in the single cells. The same variants were also seen as homoplasmic in bulk iPSC WGS data, confirming the high quality of variants in scRNAseq. In total, 28,840 heteroplasmic variants (2%<HF<=95%) we observed in 9439 cells (81.8%) were heteroplasmic or absent in their matched fibroblast cells, and 4518 heteroplasmic variants were homoplasmic in their matched fibroblast cells. A large proportion of the heteroplasmic variants (65.1%) were not seen in their matched fibroblast lines (detected HF = 0 in fibroblast cells), and only 6284 variants (21.3%) were detected in the matched fibroblast cells (Fig. 4b shows the data of 9909 cells from the three cell stages).

In order to validate the scRNAseq variant calling, we calculated the average heteroplasmy level for each variant across all the cells from the same donor (referred to as 'pseudo-bulk') (see 'Methods'). Overall, 1647 pseudo-bulk variants (HF > 0 at the pseudo-bulk level) were observed across the whole dataset (Fig. 4a). There was a high concordance between heteroplasmic level estimates from pseudo-bulk variants and WGS from the same iPSC line (scRNAseq vs WGS $P = 3.18 \times 10^{-28}$, $R^2 = 0.94$, Pearson's correlation test) (Supplementary Fig. 2e). Heteroplasmy

levels were also consistent with the variants detected in true bulk RNA-seq (scRNAseq vs bulk RNA-seq $P = 5.04 \times 10^{-29}$, $R^2 = 0.86$, Pearson's correlation test) (Supplementary Fig. 2f), further validating the variants detected by scRNAseq.

**Stable heteroplasmy in clones influence gene expression**. Next, we studied mtDNA heteroplasmy levels during differentiation at the single-cell level. We only included the cell lines where at least 20 cells were available at each stage of development to minimise sampling bias. We estimated the proportion of cells that carried the same variant within each cell line, the mean HF per variant within each cell line, and pseudo-bulk HF at all three stages (Fig. 4c, d, e). In all, 977 of the variants observed in the pseudo-bulk analysis were present in at least one of three cell stages in 24 cell lines. The single-cell analysis allowed us to compare different situations not visible in the bulk variant analysis (Fig. 5a). For example, a variant present at 50% HF in a bulk analysis could be because (1) all of the cells harboured the variant at a HF of 50%, (2) only 50% of the cells contain the variant at a HF of 100%, or (3) a mix of these two extremes (1) and (2). In our analysis, the majority of variants only occurred in a small proportion of the cells, with only 3.9, 3.8 and 3.5% of variants present in more than half of the cells in iPSC, mesendo and defendo stages (Fig. 4c).

The distribution of mean HF peaked at <2% (Fig. 4d), with a similar distribution at each developmental stage, matched by the pseudo-bulk distribution (Fig. 4e), implying a minimal change in heteroplasmy levels during differentiation. Next, we binned the proportion of cells carrying the same variant within the same line (Fig. 5b) and estimated the mutation frequency in each bin. The mutational frequencies were consistent between the three cell stages. Again, in keeping with stable heteroplasmy levels during differentiation. Finally, the HF for each individual mutation was highly correlated across all three stages (iPSC vs mesendo $P < 2.2 \times 10^{-16}$ $R^2 = 0.90$, iPSC vs defendo $P < 2.2 \times 10^{-16}$ $R^2 = 0.92$, iPSC vs mesendo $P < 2.2 \times 10^{-16}$ $R^2 = 0.88$, Pearson's correlation test) (Fig. 5c), with a tighter correlation for mutations found in a higher percentage of the cells. None of the mutations shared by >2% of cells in the iPSC population was lost during differentiation.

Given that most of the variants were rare and only seen in the minority of cells, we went on to identify discrete cell lineages within each cell line. Single cells were separated according to their origins by a Uniform Manifold Approximation Projection (UMAP)[25] based solely on the mtDNA heteroplasmic variant data (Fig. 6a), revealing their clonal origin (Fig. 6b, c, d and Supplementary Fig. 6a) (see 'Methods'). Within 27 shared variants between the original fibroblast lines and the single cells, 59.3% (16/27) of variants were also present in the vast majority (>95%) of single cells at later stages of differentiation, reflecting the ancestral lineage. Overall, 74.1% (20 of 27) variants were present >50% of single cells. There was no detectable difference between three cell stages ($P$ values >0.05, Kolmogorov–Smirnov test) (Fig. 5d). The number of variants present in the coding region and rRNAs alone allowed lineage tracing, and each cell line showed unique lineage profiles based on their heteroplasmic variants (Fig. 6 and Supplementary Fig. 6a), with similar cluster profiles observed in three cell stages (Supplementary Fig. 6b). This included multiple variants occurring in the same lineage, such as 13327G and 1392G (Supplementary Fig. 6c), which were seen in the same 20% of cells from the same donor.

Having defined sub-clones within each line, next we determined whether the mtDNA variants had an effect on function between the different sub-clones within the same cell line. First, we analysed factors contributing to the variance in gene expression for the top 4000 highly expressed genes using a linear

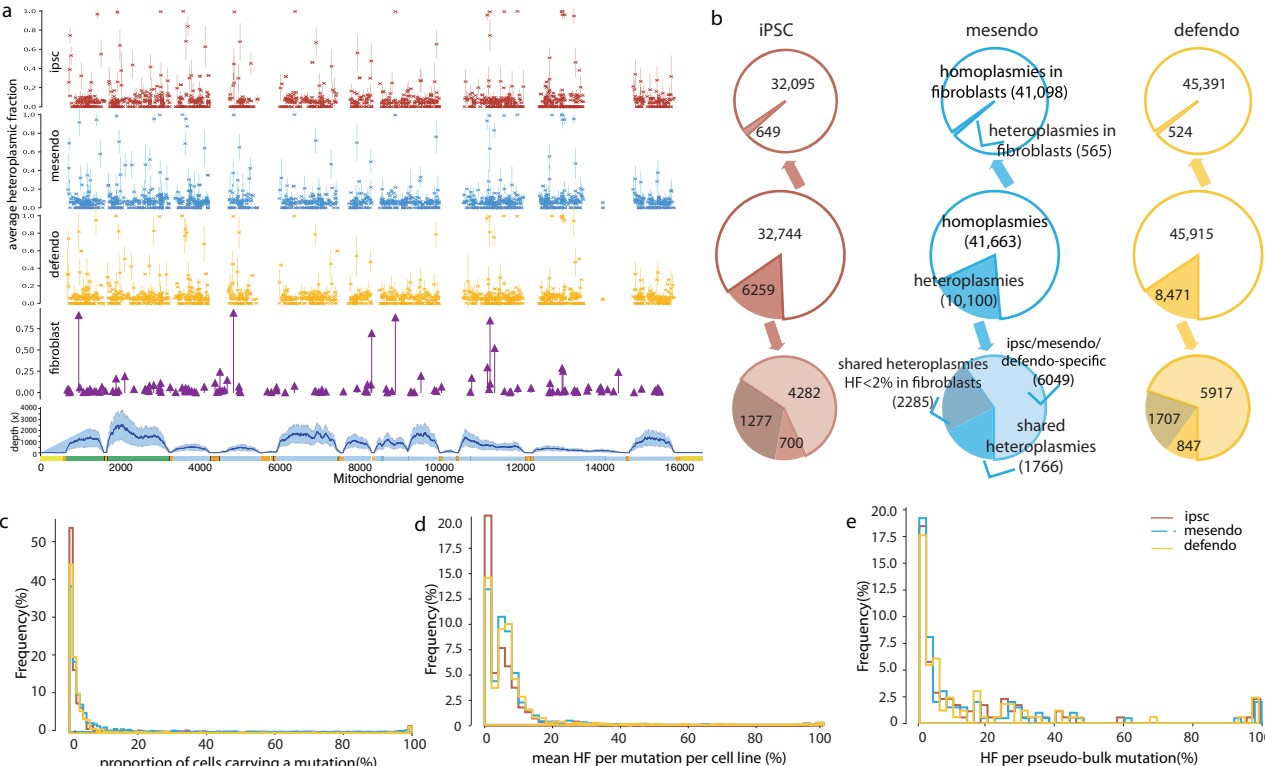

**Fig. 4 Overview of mtDNA variants detected by single-cell RNA sequencing. a** From top to bottom, mtDNA variants detected from single-cell RNA sequencing, cells defined as iPSC, mesendo and defendo stages are shown separately. Each dot represents the mean HF of each variant per cell line, and the error bar was 95% confidence interval; mtDNA variants observed in their fibroblast cells from bulk whole-genome sequencing (purple plot); mtDNA sequencing depth from single-cell RNA sequencing. Mean depth ± standard deviation (s.d.) is shown in the shaded area. Source data are provided as a Source Data file. **b** Overview of the variants detected in the single cells defined at iPSC, mesendo and defendo stages. The text labelled on mesendo stage also applies to the iPSC and defendo cell stages. **c** Distribution of the proportion of cells carrying the same variant from each cell line. The y axis shows the percentage of mtDNA variants. The majority of the variants were shared by a small proportion of cells from the same cell line. Cells defined as iPSCs, mesendo and defendo cells are shown in different colours. **d** Distribution of mean heteroplasmy fraction of each variant from each cell line. The y axis shows the percentage of mtDNA variants. The majority of the variants were low-level heteroplasmic variants (mean HF = sum HF/the number of cells carrying the same mutation). Cells defined as iPSCs, mesendo and defendo cells are shown in different colours. **e** Distribution of heteroplasmy fractions of pseudo-bulk variants from each cell line. The y axis shows the percentage of mtDNA variants. The majority of the variants were low-level heteroplasmic variants in pseudo-bulk heteroplasmy level (HF = sum HF / the total number of cells within each cell line). Cells defined as iPSCs, mesendo and defendo cells are shown in different colours.

mixed model (see 'Methods'). As seen previously, the cell developmental stage was the main source of variation, followed by the experimental batch and cell line of origin (Supplementary Fig. 6d)[3]. However, analysing the mean HF of non-synonymous mtDNA variants uncovered additional differences in gene expression attributable to the mtDNA variants. In the 12 tested cell lines, mtDNA variants accounted for >5% of the variance in expression for 897 genes in one of three developmental stages. Up to 129 genes in an individual line showed differential expression attributable to non-synonymous mtDNA variants (Fig. 5e and Supplementary Data 3), and the same 65 differentially expressed genes were observed in more than one cell line associated with the same mtDNA variants. Differential expression (DE) analysis comparing the cells carrying different mtDNA genotypes within each donor revealed 1402 DE genes (FDR < 0.1) due to 95 mtDNA variants from 17 (70.8%) cell lines (see 'Methods'). In total, 89 of 95 (93.7%) were iPSC/mesendo/defendo-specific variants observed in the single cells that were not detected in any of the matched fibroblast cells. We identified up to 291 DE genes associated with a single variant, and 52 DE genes were seen in more than one cell line (Supplementary Data 4). In conclusion, iPSC/mesendo/defendo-specific mtDNA variants defined subclones within each cell line, leading to different transcriptional

profiles. GO enrichment analysis showed that the differentially expressed genes played a key role in mitochondrial ATP synthesis, oxidative phosphorylation, epidermal cell differentiation, and telomere maintenance and organisation (FDR < 0.05) (Supplementary Data 5).

## Discussion

Human iPSCs have become a key platform for human disease modelling and therapeutic development, providing new insights into the pathophysiology, enabling high-throughput drug screening, and also providing a source for tissue-specific cell therapies. Despite showing early promise, concerns about heterogeneity between iPSC lines derived from the same donor have raised concerns about scientific reproducibility, and how generalisable the results are from the analysis of an individual cell line. Whole transcriptome and nuclear genome analysis has shown molecular heterogeneity between iPSC lines derived from the same primary cell line, contributing to the phenotypic variation[1–3], but less is known about mtDNA.

Analysing 141 iPSC and 146 primary fibroblast lines, here we confirm the age-related accumulation of mtDNA mutations in human fibroblasts, and show that cell reprogramming removes

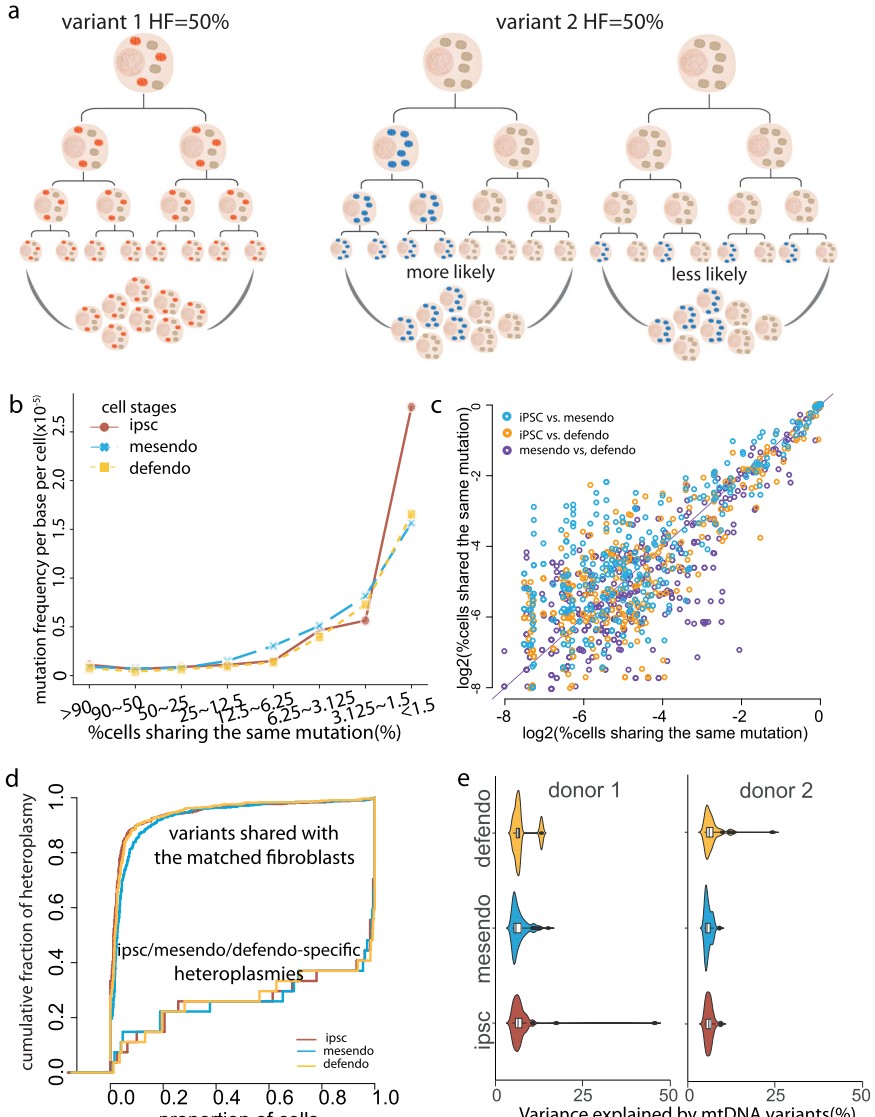

**Fig. 5 Characteristics of mtDNA heteroplasmic variants detected by single-cell RNA sequencing. a** Illustration of separate models explaining the variants with the same pseudo-bulk heteroplasmy level. A variant with 50% HF could be due to 100% of cells carrying ~50% HF heteroplasmic variants (left). Alternatively, a variant with 50% HF could be due to 50% of cells carrying homoplasmic variants in the population (right two graphs). In the middle, the variant was passed from the same ancestral cell. On the right, the variant mutated independently in a large proportion of cells. **b** The proportion of cells carrying the same variant from each cell line was grouped into different bins. Line plots show the distributions of the mutational rates estimated within each bin (where mutational rate = number of mutations within each bin from the same cell line divided by the number of cells from each cell line multiplied by 16569 (bp)). The mutational frequency profiles were consistent between the three cell stages. Cells defined as iPSCs, mesendo and defendo cells are shown in different colours. **c** Scatter plots of the log2 percentage of cells carrying the same variant from each cell line between any two of three cell stages. The HF for each individual mutation was highly correlated across all three cell stages. Source data are provided as a Source Data file**. d** Cumulative distribution of the heteroplasmic variants detected in each cell type. Variants shared with their matched fibroblast cell lines are shown in the upper left side, and iPSC/mesendo/defendo-specific variants are shown in the lower right side. **e** Violin and box plots show the percentage of the variance for gene expression explained by mtDNA variants from two independent cell lines. Cells defined as iPSCs, mesendo and defendo cells are shown in different colours.

this trend. From a mtDNA perspective, the iPSCs appear to have been 'rejuvenated', analogous to the purification of mtDNA which occurs during germline reprogramming. However, although the absolute number of mtDNA mutations in the iPSC lines was less than the matched donor fibroblasts, the mean level of heteroplasmy (heteroplasmy fraction, HF) was greater, and variants in different mtDNA regions were subject to purifying selection or were enriched through a selective advantage. We also saw iPSC-specific mtDNA mutations emerging in individual iPSC lines which displayed a distinct mutational signature, and known pathogenic mtDNA mutations reached high percentage

heteroplasmy levels. These findings are in keeping with previous observations in human embryonic stem cells (hESCs) which also harbour a wide range of different mtDNA mutations at the single-cell level[26].

It is important to note that, as with any sequencing experiment, there are technical limitations to our approach. For example, we are unable to completely exclude rare variants in the fibroblast lines that fell well below the detection threshold of 2% used in the bulk WGS, either because they were at low heteroplasmy levels in many cells, or because a very small number of cells carried homoplasmic variants. Technical limitations in the sequencing

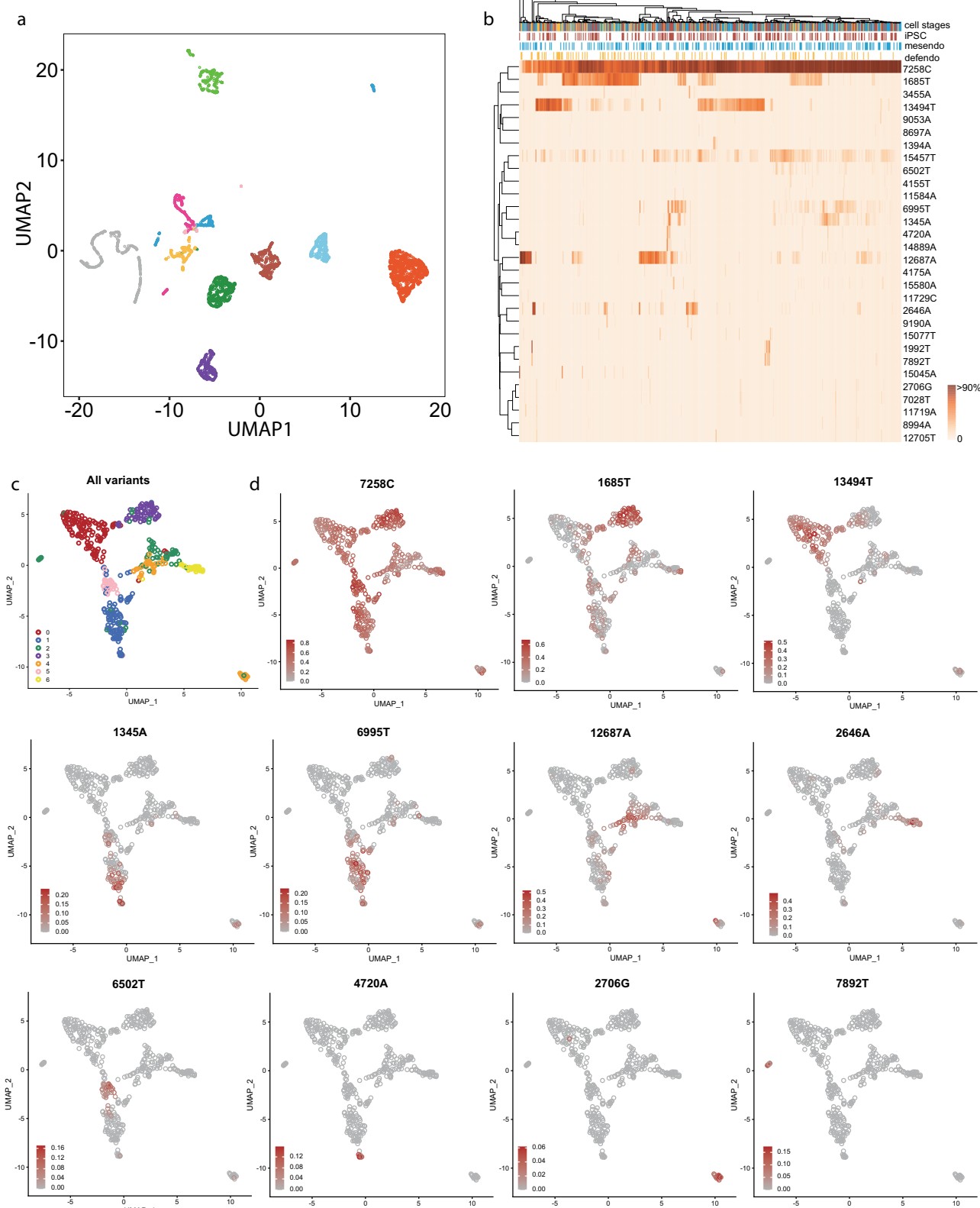

**Fig. 6 Lineage tracing using mtDNA variants reveals multiple sub-clones within each cell line. a** UMAP plot of mitochondrial mutation profiles, based on 11 cell lines with at least 300 cells. Single cells were separated according to their origins based solely on the mtDNA heteroplasmic variant data. Cells are coloured by each cell line of origin. **b** An example of hierarchical clustering by the mitochondrial genotyping (rows) for the single cells within a single-cell line. Cells are coloured by their cell stages (columns). Colour bar = heteroplasmy fraction. **c**, **d** An example of UMAP plot of mtDNA mutation profiles from a single-cell line, with cells coloured by the defined cluster (**c**), and heteroplasmy fractions of specific mutations observed in a cell line (**d**). The mutations are labelled at the top of the plots.

approach, such as the introduction of base errors, mean that we are unable to absolutely exclude the possibility of any variant present in iPSCs being present in fibroblasts, and our reported iPSC-specific mutation rate must be interpreted[13] in this context. For this reason, we have not referred to these variants as 'de novo', because we cannot be certain that these variants are not present in a very small number of molecules in the original fibroblast line. A more sensitive detection technique might reveal ultra-rare mutations in fibroblasts that are present in iPSCs at much higher levels, lowering the iPSC-specific mutation rate. However, it is also likely that a more sensitive technique would detect additional mutations in iPSCs not present in the fibroblasts, counterbalancing this effect. At this point, we can only speculate what the implications of a more sensitive detection method might be. However, this does not change our main conclusion that the heteroplasmy levels change during cell reprogramming. A re-analysis of published data[13] from human 30 iPSC clones derived from three blood samples yields a very similar mutation rate ($8.85 \times 10^{-5}$/base pair) to our result calculated from the reprogramming of human fibroblasts ($8.62 \times 10^{-5}$/base pair). This adds weight to our conclusion that reprogramming shapes the mtDNA landscape, and appears to be independent of the original cell type. However, definitive evidence will require a much more extensive comparison of iPSCs derived from different tissues from the same individual.

What mechanisms could explain the changing heteroplasmy levels we have observed during reprogramming? Rapid shifts in heteroplasmy could be due to a genetic bottleneck effect. This has been extensively studied in the female germline where the bottleneck could be due to a reduction in mtDNA content within the cell[27], packaging of mtDNA into 'segregating units' which reduced the effective population size of mtDNA molecules[28], the focal replication of mtDNA in different regions of the cell[29], and related mechanisms involving the fission and fusion of mitochondria[30]. A bottleneck effect could also occur at the cellular level due to the preferential replication or loss of cells containing specific mtDNA variants during reprogramming. This is supported by recent evidence in mice, where mtDNA variants have been shown to influence reprogramming efficiency from fibroblasts to iPSCs[19].

The selection could occur at several levels[31]. At the cellular level, differences in cellular proliferation or cell death could influence heteroplasmy levels in the whole population of cells[32]. Selection could also occur at the mitochondrial level, through increased biogenesis or selective destruction by autophagy or mitophagy[33]. Some mtDNA variants could also influence mtDNA replication itself, particularly those in the non-coding D-loop. Given emerging evidence that cell culture conditions such as the oxygen level of the addition or antioxidants can influence heteroplasmy levels[34], including during iPSCs generation[19], it is possible that the precise conditions used to generate the iPSCs influenced the heteroplasmy dynamics that we observed. However, the HipSci resource provides an opportunity to study all of these mechanisms now that the mtDNA variation has been characterised in detail.

Analysing single-cell transcriptomes at three states of differentiation allowed the identification of distinct cell lineages based on their mtDNA sequence. Unlike the reprogramming, heteroplasmy levels remained stable during the three stages of differentiation. Although only making a minor contribution to the overall variance in gene expression between donors[3], the mtDNA defined sub-clones defined within each lineage showed differences in gene expression, including key genes involved in cellular metabolism, including ATP synthesis, telomere maintenance and organisation and epidermal cell differentiation. Most of the variants (93.7%) influencing gene expression were from iPSC/

mesendo/defendo-specific variants during reprogramming. Thus, the process of cell reprogramming shapes the mtDNA landscape clone-by-clone, adding to the heterogeneity of derived iPSCs. It is highly likely that these findings will have an impact on iPSC disease models, and the use of iPSC cell lines in cellular therapies. These cell lines are available to the scientific community through the human induced pluripotent stem cell initiative (https://www.hipsci.org/)[1].

## Methods

**Donors and cell lines.** In all, 146 iPSC and 151 fibroblast lines from 151 healthy donors (male 64, female 87) were obtained from the Human Induced Pluripotent Stem Cells Initiative (HipSci, http://www.hipsci.org/)[1]. The donors' age was between 27 and 77 years (Supplementary Fig. 1a). The cell line passage number of iPSC were between passages 8 and 43 (Supplementary Fig. 1c). All donors were collected from consented research volunteers recruited from the NIHR Cambridge BioResource (http://www.cambridgebioresource.org.uk). Samples were collected initially under ethical approval for iPS cell derivation (REC 09/H0304/77, V2 04/01/2013), with later samples collected under a revised consent (REC 09/H0304/77, V3 15/03/2013).

**Generation of iPS cell lines.** The iPSCs were generated in bulk culture without physical subcloning. Details of the generation of the iPSC lines including fibroblast isolation, iPSC derivation, iPSC culture, iPSC line selection, molecular assays and cell culture for maintenance and differentiation in scRNA sequencing are available at http://www.hipsci.org/ and refs. [1,3], summarised here as follows. Primary fibroblasts were derived from 2-mm skin punch biopsies collected in advanced DMEM, 10% FBS, 1% l-glutamine, 0.007% 2-mercaptoethanol with 1% penicillin and streptomycin at room temperature. Manually dissected fragments were transferred onto a 60-mm Petri dish containing fibroblast growth medium and cultured for five days. Explants were fed every 5 days with 1 ml fibroblast medium until outgrowths appeared and were screened for mycoplasma (EZ-PCR Kit, Gene flow (41106313-001)). On reaching confluence after ~30 days, fibroblasts were passaged into 25-cm² flasks, and further passaged when 80–90% confluent into a 75-cm² flask. Cells were then expanded to confluency in 225-cm² flasks at a split ratio of 1:3, and cryopreserved at 1–2 million cells per vial in FBS and 10% DMSO or seeded immediately for reprogramming. Fibroblasts were transduced using Sendai vectors expressing human (h)OCT3/4, hSOX2, hKLF4 and hMYC51 (CytoTune, Life Technologies, A1377801) and cultured on an irradiated mouse embryonic fibroblast (MEF-CF1) feeder layer in a 10-cm² tissue-culture dish in iPSC medium consisting of advanced DMEM (Life Technologies) supplemented with 10% Knockout Serum Replacement (KOSR, Life Technologies), 2 mM l-glutamine (Life Technologies), 0.007% 2-mercaptoethanol (Sigma-Aldrich), 4 ng ml⁻¹ recombinant zebrafish fibroblast growth factor-2, and 1% pen/strep (Life Technologies). Cells with an iPS cell morphology appeared ~25–30 days after transduction. Six undifferentiated colonies per donor selected between days 30–40 were transferred onto 12-well MEF-CF1 feeder plates and cultured in iPS cell medium with daily medium change, and passaged every 5–7 days. Three of the six lines were selected based on morphological qualities (undifferentiated, roundness and colony compactness) and expanded for banking and characterisation. Each iPS cell line was passaged ~16 times before the initial molecular data for quality control, including genotyping, gene expression data, and an assessment of the pluripotency and differentiation potential of each line.

A summary of the iPSC lines used in this study is available in Supplementary Data 1.

**Extracting mitochondrial sequences from whole-genome sequencing and detecting mitochondrial variants.** Whole-genome sequencing (WGS) data of 146 iPSC and 151 fibroblast lines were obtained from HipSci (http://www.hipsci.org/). The average depth of WGS was 44-fold (s.d. = 9-fold) (Supplementary Fig. 2i). The subset of sequencing reads which aligned to the mitochondrial genome were extracted from each WGS CRAM file using Samtools[35]. We ran MToolBox (v1.1) on the resulting smaller BAM files to generate the realigned mtDNA BAM files[36]. During this process, any read pairs for which either of the two reads in the pair mapped to multiple locations were discarded, including potential nuclear mitochondrial sequences in the nuclear genome and amplification artefacts. The realigned bam files were used to call the variants and generate the vcf file for each sequence. We also used the second variant caller VarScan2[37] to call mtDNA variants from the realigned bam file (--strand-filter 1, --min-var-freq 0.001, --min-reads2 1, --min-avg-qual 30). The mpileup files used in VarScan2 were generated by Samtools with options -d 0 -q 30 -Q 30. We then filtered the variants as follows: (1) we retained variants for which the allele fractions (AFs) were above 2% from both MToolbox and VarScan2, and the allele fractions were extracted from VarScan2. The average sequencing depth of mtDNA was 1824-fold (s.d. 2249-fold), which ensured us to detect the low-level variants; (2) we retained only single-nucleotide polymorphisms (SNPs); (3) we removed variants with depth < 200×; (4) we removed variants where there were less than two reads on each strand for the

minor allele; (5) we removed variants falling within low-complexity regions (66–71, 300–316, 513–525, 3106–3107, 12418–12425 and 16182–16194).

Our initial detection threshold of 2% heteroplasmy fraction (HF) was based on our previous work[6], where we analysed samples sequenced twice with ~2000-fold depth and showed >95% reproducibility to detect a HF > 1%. Given the mean mtDNA sequencing depth of ~1000-fold in the current dataset, we set a more conservative detection threshold of 2% HF.

Mitochondrial DNA haplogroup assignment was performed using variants with allele fraction above 95% by HaploGrep2[39,40].

**Quality control of samples**. Potential DNA cross-contamination was checked using mtDNA variant calls. Two samples carrying more than ten heteroplasmic variants with similar HFs were excluded from this study. We also removed four sequences where the average depth of mtDNA was below 400×. A further four iPSCs were removed because their matched fibroblasts were unavailable or failed quality controls. After all of the sample QC steps, 146 fibroblasts and 141 iPSCs were included in the analysis. From 25 donors one iPSC line was included and from 58 donors 2 iPSC lines were included (Fig. 1a).

Although the average depth of mtDNA sequencing was clustered into two groups (Supplementary Fig. 2j), there was no detectable difference in the depth between paired fibroblasts and iPSCs (median depth in fibroblasts 777×, median depth in iPSCs 887×, $P = 0.907$, paired Wilcoxon test). Likewise, the average number of heteroplasmic variants, the heteroplasmy fraction (HF), and the average number of low-level heteroplasmic variants (HF < 5% or HF < 10%) were no detectable difference between the two clusters ($P = 0.07$, $P = 0.19$, $P = 0.10$ and $P = 0.10$, two-sided Wilcoxon rank-sum test) (Supplementary Fig. 2g, h). There was no detectable correlation between either the mean number of heteroplasmic variants or the average heteroplasmy fraction (HF) and the mtDNA sequencing depth (Supplementary Fig. 2k, l).

**Classifying the heteroplasmic variants**. We classified the heteroplasmic variants from 141 iPSCs and their matched fibroblast progenitors from 83 donors into three categories: (1) shared, if the variant was present in both fibroblasts and their derived iPSCs and was heteroplasmic in at least one of the variants; (2) lost, if the heteroplasmic variant was present in the fibroblast but not detected in their derived iPSC and (3) iPSC-specific, if the heteroplasmic variant was present in the iPSC and not detected in their fibroblast progenitor. If two iPSCs were derived from the same fibroblast, we defined them as two separate fibroblast and iPSC pairs.

Next, we defined sub-categories of heteroplasmic variants detected in the fibroblast cell lines with two derived iPSCs and their derived iPSCs: (1) shared both, if the variant was present in the fibroblast and both derived iPSCs and was heteroplasmic in at least one of the variants; (2) shared one/lost one, if the variant was present in the fibroblast and only one of two iPSCs and was heteroplasmic in at least one of the variants; (3) iPSC-specific both, if the same heteroplasmic variant was present in both iPSCs and not detected in their matched fibroblast; (4) iPSC-specific one, if the heteroplasmic variant was present in only one of two iPSCs and not detected in their matched fibroblast; (5) lost both, if the heteroplasmic variant was present in the fibroblast but not detected in any of their derived iPSCs.

**Analysing age correlation**. To understand the relationship between donor age and the mean number of heteroplasmic variant and average heteroplasmy fraction in fibroblast cell lines and iPSC lines, we applied linear regression to each dataset using R[41].

$$Model\ 1 < - \ lm(Nfibro \sim a + Sex + Hap) \quad (1)$$

$$Model\ 2 < - \ lm(Nipsc \sim a + Sex + Hap) \quad (2)$$

$$Model\ 3 < - \ lm(HFfibro \sim a + Sex + Hap) \quad (3)$$

$$Model\ 4 < - \ lm(HFipsc \sim a + Sex + Hap) \quad (4)$$

where Nfibro and Nipsc are the mean number of heteroplasmic variant from each sample, a is the donor age, HFfibro and HFipsc are the logit transformed average heteroplasmy fraction from each sample, Sex is the sex and Hap is the mac-haplogroup from each donor.

**Predicting the direction of shifts of heteroplasmic variants**. To understand whether multiple factors affect the heteroplasmic shifts between fibroblast cell lines and iPSC lines, logistic regression was applied using R.

$$Model5 < - \ glm(S \sim HFvfibro + Rmt + Hap + Age, family = binomial) \quad (5)$$

where $S = 1$ if the normalised heteroplasmic shift was above 0, and $S = 0$ if normalised heteroplasmic shift was below 0, HFvfibro was the logit transformed heteroplasmy fraction from each variant in the fibroblast cell lines, Rmt was the mtDNA regions (D-loop, non-synonymous, synonymous, rRNA and tRNA variants), Hap was the macro-haplogroup from each donor, and age was the donor age.

**Defining known pathogenic mutations**. The initial list of pathogenic mutations was from MitoMap (Disease Mutations with mutation status as 'Cfrm')[42] and the previous study[38]. We then reviewed each variant by searching online literatures. Based on previously published criteria[43], a total of 87 single-nucleotide mutations were included in the final list of pathogenic mutations[44].

**Analysing mutational signature and strand bias**. Mutational spectra were derived from the revised Cambridge Reference Sequence (rCRS) and alternative alleles at each variant site. The resulting spectra were composed of the six distinguishable point mutations (C:G > T:A, T:A > C:G, C:G > A:T, C:G > G:C, T:A > A:T and T:A > G:C). Each signature was displayed using a 96 substitution classification defined by the substitution class and the sequence context immediately 3' and 5' of the mutated base[20,24]. The substitution rate for each trinucleotide context was estimated by the number of substitutions divided by the frequency of the trinucleotide context present in the mtDNA reference genome (rCRS), in the light (L) and heavy (H) strands. The signature profiles from the total heteroplasmic variants observed in fibroblasts and iPSCs, lost heteroplasmic variants in fibroblasts, iPSC-specific heteroplasmic variants in iPSCs and shared heteroplasmic variants between fibroblasts and iPSCs were considered. To compare the signature profiles between the different mtDNA regions, we estimated the frequency of six point mutations in the L and H strands. To explore the mechanism of mutagenesis we correlated the signature profiles of mtDNA substitutions from fibroblasts, iPSCs, shared and iPSC-specific heteroplasmic variants with 30 cancer-specific mutational signatures in the nuclear DNA identified[24].

**Estimating relative mtDNA copy numbers from WGS**. We estimated the relative mtDNA copy number from WGS by comparing the mean read depth of the autosomes to the mean depth of mtDNA. Mean autosomal depth (DPautosome) was calculated from 2.685 Gb autosome regions (without chromosome gaps). And mean mitochondrial depth (DPmt) was obtained from the full length of mtDNA genome (16,569 bp). Relative mtDNA copy number (CNmt) was calculated in a diploid cell as below:

$$CNmt = 2 \times DPmt/DPautosome \quad (6)$$

**Detecting mitochondrial variants from bulk RNA sequencing**. Bulk RNA-seq FASTQ files for 332 iPSC lines were obtained from the ENA project: ERP007111 and EGA projects: EGAS00001000593. Raw sequencing data were used to detect mitochondrial variants using the same pipeline as described above. Variants with low-quality RNA-seq were discarded based on the criteria applied in WGS data. In addition, we removed the highly heteroplasmic variants that were specific to mtRNA (2617 A > G, 2129 G > A, 295 C > T, 5746 G > A, 13710 A > G and 5985 G > T), including three variants (295 C > T, 13710 A > G, and 2617 A > G) previously reported as artefacts (Bar-Yaacov et al., 2013). 102 of 332 iPSC lines that had matched WGS data available were included in this study[45].

**Detecting mitochondrial variants from single-cell RNA sequencing**. Single-cell RNA-seq FASTQ files for 36,044 cells were obtained from the ENA project: ERP016000 and EGA project: EGAS00001002278 and EGAD00001005741. Processed single-cell count data were obtained from Zenodo: https://zenodo.org/record/3625024#.Xil-0y2cZ0s. The details of cell culture and single-cell sequencing were previously described[3]. The pipeline developed for the WGS data was used to detect mtDNA variants from raw FASTQ files, with several additional filtering steps: (1) retaining the cells with mtDNA genome sequencing depth above 200×, (2) excluding the iPSC/mesendo/defendo-specific variants only detected in a single cell from the same cell line which was more likely from sequencing error, (3) defining heteroplasmic variants with HF between 2% and 95%, due to lower sequencing depth in single-cell RNA-sequencing data compared to WGS, (4) excluding the cells with >20 heteroplasmic variants, (5) retaining the mtDNA regions with mean sequencing depth above 200× (11,704 bp from 2463 bp of rRNAs and 9241 bp of coding regions). This resulted in 11, 538 cells from 60 cell lines which also had WGS data available from their matched fibroblast cell lines.

**Defining developmental stages of the single cells**. The pseudotime of the single cells from single-cell RNA-seq was obtained from Zenodo: https://zenodo.org/record/3625024#.Xil-0y2cZ0s. The stage labels post iPSC (mesendo and defendo) were defined using the methods previously described[3]. Briefly, cells were assigned to the mesendo stage if they were collected at day 1 or day 2 and had pseudotime values between 0.15 and 0.5. Cells were assigned to the defendo stage if they were collected at day 2 or day 3 and had pseudotime values higher than 0.7. Cells with intermediate pseudotime (between 0.5 and 0.7) were assigned as undefined. Overall, we assigned 9969 (86.4%) cells to any of the stages (iPSC, mesendo and defendo).

To compare the variant profiles between three cell stages, we included the cell lines where at least 20 cells were available at each stage of development to minimise sampling bias, which included 6881 cells from 24 cell lines. The average allele frequency of each variant was estimated across all the cells from the same cell line at each cell stage, which we termed pseudo-bulk variants.

**Defining cell lineages**. For the analysis of the cell lineages, we only included 692 variants detected in more than 1% cells within the same cell line from 24 cell lines which had >20 cells assigned to each cell stage. In all, 27 of 692 variants were also present in the matched fibroblast cell lines, and 665 were iPSC/mesendo/defendo-specific variants in the single cells. We performed a Uniform Manifold Approximation and Projection (UMAP)[25] of the cells from the different cell lines and within each cell line using the mitochondrial DNA variants described above. UMAP was analysed using the UMAP package[46] with default parameters in R and visualised using the M3C package[47] in R.

**Variance component and differential expression (DE) analysis**. We only included the cells and variants meeting the following filtering criteria: (1) cells were assigned to one of three cell stages (iPSC, mesendo and defendo), (2) cell lines with at least 100 cells and >20 cells assigned to each cell stage and (3) retaining variants where >15 cells carried one of two alleles in each cell stage (Supplementary Data 1). To perform variance component analysis for each cell line, we only included 12 cell lines with more than 100 cells from each cell stage. We identified the top 4000 most variable genes using a linear mixed model, and genes were ranked by the significance of deviation[48]. Variance component analysis was performed on top 4000 most variable genes using a linear mixed model in the variancePartition R package[49]. We included the cell stage, the experiment and the cell line identity as random effects. To estimate the effects of mtDNA variants on gene expression, we identified top 4000 most variable genes in the cells from the same cell stage, the same experiment and the same cell line identity. Variance component analysis was performed by partitioning gene expression variation into mean heteroplasmy fractions of mtDNA non-synonymous variants and residual noise.

To further understand the effect of each mtDNA variant on gene expression, we tested the differences in gene expression between cells carrying different genotypes of mtDNA variants from the same cell line. We fitted a generalised linear model for the gene expression data within each cell line from the same experiment with the cell stages as covariates using MAST R package[50]. mtDNA genes and ribosomal genes were excluded from the analysis. Gene Ontology (GO) enrichment analysis was performed using Enrichr[51–53].

**Statistical analysis and plotting**. All statistical analyses in this study were performed using R[41] and Python (http://www.python.org). Figures were generated using Matplotlib (https://matplotlib.org) in Python and R. Circos plots were made using Circos[54].

**Reporting summary**. Further information on research design is available in the Nature Research Reporting Summary linked to this article.

## Data availability

All HipSci data used in this study can be accessed through http://www.hipsci.org. Whole-genome sequencing data are available under accession numbers: ERP017015. Bulk RNA-seq data are available under accession numbers: ERP007111 and EGAS00001000593. Single-cell RNA-seq data are available under the accession numbers ERP016000 and EGAS00001002278, EGAD00001005741. Processed single-cell count data are downloaded from Zenodo: https://zenodo.org/record/3625024#.Xil-0y2cZ0s. Source data are provided with this paper.

## Code availability

Code is available at https://github.com/WeiWei060512/HipSci_mtDNA_paper. https://doi.org/10.5281/zenodo.5136677.

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

## Acknowledgements

The fibroblast cell lines within the HipSci consortium were collected from participants of the National Institute for Health BioResource. P.F.C. is a Wellcome Trust Principal Research Fellow (212219/Z/18/Z), and a UK NIHR Senior Investigator, who receives support from the Medical Research Council Mitochondrial Biology Unit (MC_UU_00015/9), the Medical Research Council (MRC) International Centre for Genomic Medicine in Neuromuscular Disease (MR/S005021/1), the Leverhulme Trust (RPG-2018-408), an MRC research grant (MR/S035699/1), an Alzheimer's Society Project Grant (AS-PG-18b-022) and the National Institute for Health Research (NIHR) Biomedical Research Centre based at Cambridge University Hospitals NHS Foundation Trust and the University of Cambridge. The views expressed are those of the author(s) and not necessarily those of the NHS, the NIHR or the Department of Health and Social Care.

## Author contributions

P.F.C. conceived the study. W.W. performed the analysis. D.J.G. provided critical oversight. P.F.C., W.W. and D.J.G. wrote the paper.

## Competing interests

The authors declare no competing interests.
