## [Peer Review File · Nature Communications]

REVIEWER COMMENTS

Reviewer #1 (Remarks to the Author):

In the manuscript "Cell reprogramming shapes the mitochondrial DNA landscape" by Wei Wei et al, the authors use whole genome sequencing data from the human induced pluripotent stem cell initiative to examine the impact of reprogramming on mutations on the mitochondrial genome (mtDNA) and the potential consequences of these mutations on transcriptional heterogeneity in differentiating iPS cells. They conclude that reprogramming induces de novo mutations, thereby impacting on the mutational landscape. Furthermore, they use single cell RNA-seq to show that mutations in mitochondrial DNA affect gene expression patterns in differentiating iPS cells. Because the authors were able to use the data from the HiPSCi consortium, this study represents the largest study on mitochondrial mutations in iPSCs of its kind. In my opinion, this merit of the study is also its pitfall, because the experimental set-up was not designed to address the key what the mutational impact of cellular reprogramming on the mtDNA landscape is. While the study may be of potential interest, I am of the opinion that several important matters need to be addressed before it is suitable for publication in Nature Communications.

Major comments:

mtDNA sequencing depth was nearly 2000 fold, which sounds very high, but in fact isn't if you consider that each cell contains around 1000 mitochondria, which means that the equivalent that was sequenced is the mitochondrial content of just two cells. As a consequence, it is impossible to accurately capture the full heterogeneity of mtDNA variants in the large bulk population of cells that is routinely used as input for WGS. Because sequencing was performed on non-clonal lines, many mtDNA variants are missed, namely those variants with a low heteroplasmy fraction, but also those that are present in a small fraction of the cells, but at a high heteroplasmy fraction. Moreover, it is unknown for any given mtDNA variant if it is shared by all the cells or only by a proportion of the cells. As a consequence, it is very difficult to conclude whether mutations detected in iPSCs, but not in fibroblasts are de novo mutations or were already present below the detection limit in the fibroblast population and have become detectable as a result of the reprogramming bottleneck. This also means that conclusions on the de novo mutation rates cannot be drawn from the current study. A different experimental set-up using clonal steps would fit that purpose better. To capture all the mtDNA variants prior to reprogramming, a clonal fibroblast line is necessary to avoid the large heterogeneity between cells in the mitochondrial mutations. A second clonal step of the derivative iPS cell line is required to capture all the mitochondrial mutations after reprogramming. In my opinion, this is an essential experiment that is lacking from the current manuscript.

For people outside the mitochondria field, the study is difficult to understand. The authors also provide very limited insight into the possible underlying mechanisms. Place results in the right context and more extensively discuss the possible reasons for the findings, including selection at the mitochondrial level, selection at the cellular level, reprogramming induced changes in the balance between DNA damage and repair, reversal of mutations, induction of mutations, unequal segregation, heterogeneity in cell types of the starting population and the possibility of cell type specific patterns with possibly some cell types with specific mutational patterns being reprogrammed more efficiently etc.

Essential explanation of certain methods is lacking and makes this study difficult to replicate. In general, the authors tend to describe what they did, not how the analysis was performed. Which R-packages were used? Relevant information such as how iPS cells were derived (sendai, mrna, lentiviral...), culture conditions (low oxygen/normoxia), how DNA was isolated, library prep, sequencing etc is lacking. The authors now refer to the HiPSCi website, but this is not a preferred

method to describe these methods, because it doesn't encourage readers to take along this relevant information for accurate interpretation. Moreover, after publication, the paper may outlive the website.

Minor comments

All mutations detected in iPSCs, but not in fibroblasts should not be called "de novo" unless solid evidence is provided these were indeed absent in the parental population.

In line with this: the increase in mean HF per iPS cell line suggest mtDNA variants have become detectable as a result of the reprogramming bottleneck.

The iPS cell lines were not clonally expanded and should therefore not be referred to as clones throughout the manuscript.

The iPS cell lines were sequenced at highly different passages (8-43). I expect the late passage lines two have become more clonal due to drift or selection. This may influence HF detection. Please comment and discuss in manuscript.

In the results and also the M&M section (phrased a little differently) the authors mention that "at least one iPSC line from 83 donors (56.9%) and 2 iPSC lines from 58 donors (39.7%)" were included. This is misleading, because now it seems there were $(83*1) + (58*2) = 193$ donors. Rephrase, for example: From 25 donors one iPSC line was included and from 58 donors 2 iPS cell lines were included.

Replace "reference sequence (rCRS)" with "revised Cambridge Reference Sequence (rCRS)".

Mention the absolute numbers of heteroplasmic mtDNA variants found in fibroblasts and in iPSCs in the results section, just prior to the mean number of variants.

How are the mutation frequencies (mutation rates per base pair per genome) calculated? Were the number of mitochondria per cell taken into account (I think they should), and if so, how were these determined?

Could the authors justify the use of Stouffer's method? The P-value from this method doesn't seem to reflect the few more C>A, C>G, and T>A variants in iPS cells. Since the trinucleotide context is not taken into account in their conclusions, the authors could also use a Pearson's chi square test on mutation spectra instead.

Why were indels not analysed, also because these are relevant for the mismatch repair signature that seems to be present in the mitochondria of iPS cells?

Page 18: replace "36,044 cell lines" with "36,044 cells"

Page 31 figure 4b is unclear. Enlarge and add absolute values

FigS1A increase bar sizes of histograms to fill gaps

Page 23 ref 4 is incomplete

Reviewer #2 (Remarks to the Author):

We et al. present a very interesting manuscript where they analysed the landscape of mtDNA mutations detectable in human fibroblast, its fate upon reprogramming to iPS cells and their further fate upon differentiation. They investigate a number of interesting correlations as the evolution of the mtDNA mutation landscape with the age of the donors, the impact of reprogramming in the heteroplasmic score, the lost and/or gain of mutations, etc. The analysis performed is sound and well performed. The conclusions are very relevant in several aspects of the evolution of mtDNA heteroplasmy in humans but also to understand the impact of mtDNA mutagenicity in the use of human iPS cells. Among the numerous observations provided they concluded:

- 1) Most age-related fibroblast mtDNA mutations are lost during reprogramming.
- 2) During reprogramming novo mtDNA mutations appear in higher proportion of mtDNA molecules, favouring non-synonymous protein-coding and tRNA variants, including known disease-causing mutations.
- 3) The mtDNA mutations affect ROS species production, amino acid metabolism and cell growth through indirect effects on nuclear gene transcription.

I do not have any major concern regarding the manuscript but some suggestions that may be considered by the authors that could help to improve or extend the implications of their study:

- 1) At the end of the second paragraph of the introduction the authors explain that mtDNA mutations influence the transcriptomic profile of the cells and quoted two papers that focused only in tumours. However this influence is also apparent in healthy cells in an organism as it was described in mice (PMID: 32832682; PMID: 27383793).
- 2) At the end of the first paragraph in Results section the authors state: "We then determined frequency distribution of major population-specific mtDNA haplogroups (macro-haplogroups), confirming that the vast majority (98%) of donors were representative of the UK population (Figure S1B). As a further QC step, we only included heteroplasmic variants with bulk heteroplasmy allele fractions (HFs) between 2% and 98%"
Mitochondrial haplogroups U and K are very often named as UK haplogroup. However, UK does not mean United Kingdom when referring to mtDNA haplogroups. I suggest that to avoid confusion UK population here would better be expressed as United Kingdom population.
- 3) Figure 2 B, C, E and F represent linear regression models. The statistical significance and the quality of the model of correlation estimated should be included also in the figure. In this respect it seems clear that the linear regression model does not explain well the relation between the compared variable in those graphs even when the correlation is significant. I believe that this needs to be discussed.
- 4) Equally, the segmentation in D-loop, rRNA, tRNA and Protein encoded (synonymous vs non-synonymous) is natural but given the amount of data and because the structure of the rRNAs, tRNAs and the 13 proteins encoded is known additional relevant information can be retrieved. In the rRNA and tRNA observed mutations: is there any difference between double vs single strand structural areas? Within the proteins coding genes: Is there any difference in the protein according to the complex where they belong (CI; CIII; CIV; CV)?
- 5) Regarding the mechanism of generation of de novo mutations upon iPS reprogramming the authors concluded that "To determine whether some of the de novo mutations were actually present in the fibroblasts, but below our detection threshold (HF<2%), we re-analysed the fibroblast data without a lower cut-off filter for HFs. This showed that a minority (14.8%; 40 / 270) of the previously classified de novo variants were actually present in the original fibroblast line at HF<2%. Factoring this into account, the de novo mutation rate was 8.62×10^{-5} per base pair per genome per reprogramming,

which is a conservative estimate based on mutations with a HF \geq 2% in the iPSCs and a HF $>$ 0 in their matched fibroblast cell lines.”

The reduction of the cut-off below 2% allow to detect that 15% of the mutations considered de novo should be reclassified as pre-existing. This is a clear warning regarding the origin of all de novo classified mutations since the experimental approach is not capable to guaranty that all pre-existing mutations could be detected regardless their initial frequency since under a given threshold it would be not possible to determine errors vs true pre-existing mutations. Which is the threshold cut-off that you estimate that can be unequivocally identify a pre-existing mutation? Which is the probability that a pre-existing mutation would remain undetectable? Those are interesting estimations since your analysis raised the question of the mechanism of de novo generation mutations and if the mtDNA replication fidelity is relaxed in a programmed fashion at any stage of pluripotency. Would you comment on these issues?

6) Regarding the mechanism of mutation selection and generation upon reprogramming the manuscript does not explain the culture conditions of the cells nor that of the reprogramming that may be of critical relevance for the interpretation of the observations. Thus, it is known that in conventional culture conditions cells are maintain in 20% oxygen while in the body the oxygen concentration does not goes over 5%. It is also described that low oxygen concentration improve significantly genetic stability (PMID: 22139129) and the efficiency of iPS formation from fibroblast when performed at 4% Oxygen vs 20% Oxygen (PMID: 31588014) the distribution of heteroplasmic mtDNA between fibroblast and iPS cells if performed and that culture in the presence of antioxidants prevents the shift in heteroplasmy between fibroblast and derived iPS cells (PMID: 31588014). Given the warning message derived of this work it would be of interest commenting on these issues.

7) The discussion between “genetic bottleneck” and “cellular bottleneck” is quite interesting and is, as suggested by the reviewers related with the efficiency of reprogramming. In this respect there are strong evidence that demonstrate that the mtDNA complement determine efficiency of reprogramming from mouse fibroblast to iPS cells (PMID: 31588014).

8) This reviewer has problem to understand the dichotomic criteria of classification of the mutations in iPSC as increasing HF or decreasing HF without considering a third category of non-sifting. In fact, I would find that a high number of them would be better classified in this category. Which is the proportion of shifting that guaranty that a significant HF shift has been produced? I strongly believe that this data should be revisited considering the three categories. Moreover, why the authors considered that the fact the behaviour of HF for shared heteroplasmic variants between iPSCs and fibroblast argue in favour of a genetic bottleneck vs a cellular bottleneck?

9) The authors notice that: “The preferential loss of D-loop variants and propagation of non-synonymous and tRNA variants are the complete opposite of what is seen during germ-line transmission”. I do not understand why they manifest surprise for that since reprogramming imply inverting the direction from germ-line mtDNA transmission and it has been described that pluripotent cells modulate heteroplasmy differently than mouse embryonic fibroblast from and that reprogramming the MEFs to iPS recapitulate the pluripotent cell behaviour (PMID: 31588014).

Reviewer #3 (Remarks to the Author):

The authors perform a deep analysis comparing mtDNA sequences in fibroblasts and iPSCs. They find a wide range of interesting results, including that reprogramming appears to have diverse effects on the presence of different variants in cellular mtDNA populations.

This is an important and timely result for several reasons -- the broader mystery of heterogeneity in iPSCs and the specific usage of iPSCs in mitochondrial investigations, both of which are informed by

this work. I greatly enjoyed this paper, which is clearly written (notably so, given the amount of information it conveys), with clear analyses, and with data presented well in the figures (I am particularly grateful for the distributional detail with which important quantities are plotted). I have a set of minor points but only four larger-scale ones (marked with ** below, and summarised here) before recommending acceptance.

1. Although most methodological choices are described in admirable detail, there is one quantity with potentially high influence that is less clear -- the lower cutoff used when characterising variants as arising de novo vs present in the original. Some more discussion and investigation of the influence of this choice would be useful.
2. The argument about a cellular bottleneck would seem to warrant more support. I believe the story but at the moment the copy number measurements don't necessarily support the argument in the way that is discussed.
3. I couldn't find a link to the code for the authors' pipeline to reproduce the analysis and figures. This should be provided, or a statement should be made describing why it can't be.
4. There are several instances where parametric tests (often t-tests) are used when their assumptions (often normality) are clearly not met by the data. These should be replaced with non-parametric alternatives (in most cases it is clear from the data that this will not greatly affect the results). Also I request rewording several instances of "no correlation", "no effect" etc to "no detectable correlation", "no detectable effect" and similar, reflecting the fact that absence cannot be claimed based on a high p-value.

Iain Johnston

l39 -- "similar experimental outcomes" -- it's not clear whether this means that (i) a similar amount of phenotypic difference is observed between experiments, or that (ii) phenotypes are similar between experiments (i.e. less difference). can this be clarified?

l63 -- "extreme mtDNA diversity" -- a summary comparison would be useful here. what magnitude of what measure of diversity, compared to what we see physiologically in humans?

l79 -- an odd test. are you using $p = 0.05$ to support an absence of a difference? it would be best to give the actual depth numbers here so the reader can compare for themselves.

l81 -- expanding and citing rCRS would help for non-experts

l87 -- Fig S1B does not by itself support this claim that the sample is representative of the UK. I believe it from looking at the haplogroup makeup, but please provide a population distribution or other evidence for comparison, and explain where the 98% figure comes from.

l100 -- please replace "a greater number" with "a great mean number" as -- if I am understanding -- your regression model looks at mean number, not number

l102 -- the test here seems different from the claim being made. from Fig 2C it indeed looks like D-loop heteroplasmy increases with age, but so does heteroplasmy in other regions, and you haven't explicitly tested whether the D-loop increase is steeper than others? of course "largely" could mean anything but the current phrasing implies that the D-loop is dominant. can you either demonstrate this, or rephrase e.g. "due in part to an increase..."

l112 -- quick typo Figures -> Figure

l117 -- "Ultimately" and "initially" contrast a bit confusingly here. could the "ultimately" be omitted?

l130 -- quick typo "analysis *of* high-depth"

l131 (and l75) -- did I miss why these 83 were chosen in particular? any risk of bias here?

l134-137 -- "no correlation", "this was not significant" -- please replace with "no detectable correlation" and "our observations did not provide statistical support for this trend". It looks to me as if there actually might be a trend underlying the fractions in Fig S3A; and in general of course we can't claim that there is no correlation just because we have a high p-value!

l140 -- I have no doubt that this is a true result but a t-test is not appropriate here as the distributions are clearly not normal. Please replace with a non-parametric test for robustness.

l150 -- is this pointing to the correct SI figure? I couldn't see mutation counts in S1C? also as above please replace l148 "no correlation" with "no detectable correlation"

l173 -- quick (possible?) typo -- fibroblasts -> fibroblast

** l185 -- this approach would seem to warrant more description. what was the new cut-off you used? if you used a yet lower one, would you see that still more variants were actually present in the original lines? what is the tradeoff in terms of false positives or other issues -- ie why not use the lowest conceivable cutoff? this would seem to have a major influence on what is classified as de novo and what is transmitted.

** l198-200 -- I'm not really convinced by this. the fact that you see higher copy numbers "after" than "before" doesn't really support the presence of a reprogramming bottleneck -- it just reflects how things are in fibroblasts. there could be an argument that the (random) expansion of mtDNA from fibroblast to iPSC contributes genetic variance (see e.g. reamplification models in <https://www.frontiersin.org/articles/10.3389/fcell.2019.00294/full>) -- but the differences in copy number look too small for this to be the case. I'm an advocate of random turnover contributing genetic variation, which could apply here, as could a (not mutually exclusive) variant of the Wai-Teoli-Shoubridge picture <https://www.nature.com/articles/ng.258?foxtrotcallback=true> where only a subset of mtDNAs are "allowed" to replicate. those processes can generate genetic variability (a genetic bottleneck) without necessitating copy number reduction (a physical bottleneck). It will be clear from my comments (and my signature) that I am not a stem cell expert! Does the absence of physical sub-cloning mean that these iPSCs are not generated / maintained through repeated cell divisions? If so, could you say this explicitly, to help readers like me? If not -- random partitioning of mtDNA at cell divisions during iPSC generation / maintenance is another potential source of genetic variance.

l200 -- a t-test doesn't seem appropriate here given how non-normal the distributions are. please replace with a non-parametric alternative.

l201 -- quick typo -- precise -> precisely

l221 -- a t-test doesn't seem appropriate here given how non-normal the distributions are. please replace with a non-parametric alternative.

l224 -- not quite sure what "completely out with" means here

l227-233 -- When you say "HS decrease", I think this means that HS were more likely to be negative, meaning a decrease of heteroplasmic fraction? At the moment it reads like the shifts were lower in magnitude -- ie a smaller change. Same for "increase".

l293 -- quick typo -- citation styled inconsistently with other citations

l316 -- how can your definition of homoplasmy be $HF > 95\%$ if you are detecting and analysing variants between your lower cutoff of 2% and 5%?

l325 -- "HF=0" -- should this be "detected HF=0" or "HF < [cutoff]"?

l341 -- quick typo -- "estimated *the* proportion"

l352 -- quick typo -- "at a" -> "at"

l371 -- please replace "no difference" with "no detectable difference"

l374 -- quick typo -- "lineage profile*s*"

l384 -- I like this analysis but it must be noted that the amount of variance explained by all of these features is pretty low (the residuals are correspondingly pretty high). Can you give a ballpark for this in the main text?

l392 -- $FDR < 0.1$ seems quite permissive given that you've used $p = 0.05$ to imply absence of an effect elsewhere. is there a reason for this choice?

l484 -- quick typo -- "Figures" -> "Figure"

l490 -- please replace "no different" with "not detectably different"

l491 -- please replace "no correlation" with "no detectable correlation"

l517 -- there is an official citation for R, which the initiative prefers you to use

l618 -- please cite the UMAP algorithm (I think it's an arxiv preprint)

** l648 -- I couldn't find a link to code to reproduce the analysis and figures of the paper. Please provide this (or state why it is not available).

Response to reviewers

Reviewer #1

Reviewer: mtDNA sequencing depth was nearly 2000 fold, which sounds very high, but in fact isn't if you consider that each cell contains around 1000 mitochondria, which means that the equivalent that was sequenced is the mitochondrial content of just two cells. As a consequence, it is impossible to accurately capture the full heterogeneity of mtDNA variants in the large bulk population of cells that is routinely used as input for WGS. Because sequencing was performed on non-clonal lines, many mtDNA variants are missed, namely those variants with a low heteroplasmy fraction, but also those that are present in a small fraction of the cells, but at a high heteroplasmy fraction. Moreover, it is unknown for any given mtDNA variant if it is shared by all the cells or only by a proportion of the cells. As a consequence, it is very difficult to conclude whether mutations detected in iPSCs, but not in fibroblasts are *de novo* mutations or were already present below the detection limit in the fibroblast population and have become detectable as a result of the reprogramming bottleneck. This also means that conclusions on the *de novo* mutation rates cannot be drawn from the current study. A different experimental set-up using clonal steps would fit that purpose better. To capture all the mtDNA variants prior to reprogramming, a clonal fibroblast line is necessary to avoid the large heterogeneity between cells in the mitochondrial mutations. A second clonal step of the derivative iPSC cell line is required to capture all the mitochondrial mutations after reprogramming. In my opinion, this is an essential experiment that is lacking from the current manuscript.

Response: The reviewer refers to the depth of whole-genome sequencing in the bulk fibroblast and iPSC lines. In bulk sequencing, DNA from all of the cells is extracted and a random sample of the DNA templates is sequenced. A sequencing depth of ~2000-fold allows the confident sampling of any variant present at >2% variant allele frequency (VAF) in the original sample. Importantly, exactly the same method was used for the fibroblasts and the iPSCs, and we were very explicit about the detection threshold and our method.

We appreciate the reviewers concern that any variant in the iPSCs could, in theory, be present in the fibroblasts at a very low level (including variants at a high % in rare single cells). Taken to its extreme, a mutation could affect just one mtDNA molecule in a pool of a million cells (or 1 in 10^{10} molecules). Confidently ruling out this situation is extremely challenging because base-errors in any sequencing technique occur at a much higher frequency than the theoretically undetected mutations. It is therefore impossible to completely exclude the reviewers concern (which is based on an un-testable hypothesis).

Any experimental approach needs to clearly define the inevitable technical limitations. We believe that we have done this, explicitly reporting *de novo* mutations in iPSCs as variants not detected in fibroblasts using this approach. We also discussed the implications of using a different detection threshold on page 7, para 1.

The reviewer suggests that we repeat all the experiment by cloning single fibroblasts, but unfortunately even this would not address the reviewer's concern. It is not possible to sequence mtDNA in living cells, so how would we ever know what the starting population of mtDNA was in the original single cell? Once a cell divides, the heteroplasmy levels will change, moving above and below any experimental detection threshold creating exactly the same theoretical problem that the reviewer points out. It is impossible to start with a 'tabula rasa' and then study the downstream consequences of re-programming. This inevitably means some compromise.

Our approach was to look at very high numbers of cell lines and cells, and be explicit about our methods and their limitations, and to interpret our findings in this context. This approach has satisfied reviewers 2 and 3.

However, to address the specific concerns of reviewer 1 we have discussed these specific points in the discussion:

Page 14. It is important to note that, as with any sequencing experiment, there are technical limitations with our approach. For example, we are unable to completely exclude rare variants in the fibroblast lines that fell well below the detection threshold of 2% used in the bulk WGS, either because they are at low heteroplasmy levels in many cells, or because a very small number of cells carried homoplasmic variants. Technical limitations in the sequencing approach, such as the introduction of base-errors, mean that we are unable to absolutely exclude the possibility of any variant present in iPSCs being present in fibroblasts, and our reported de novo mutation rate must be interpreted in this context. A more sensitive detection technique might reveal ultra-rare mutations in fibroblasts that are present in iPSCs at much higher levels, lowering the de novo mutation rate. However, it is also likely that a more sensitive technique would detect additional mutations in iPSCs not present in the fibroblasts, counterbalancing this effect. At this point, we can only speculate what the implications of a more sensitive detection method might be. However, this does not change our main conclusion that the heteroplasmy levels change during cell reprogramming.

Reviewer: For people outside the mitochondria field, the study is difficult to understand. The authors also provide very limited insight into the possible underlying mechanisms. Place results in the right context and more extensively discuss the possible reasons for the findings, including selection at the mitochondrial level, selection at the cellular level, reprogramming induced changes in the balance between DNA damage and repair, reversal of mutations, induction of mutations, unequal segregation, heterogeneity in cell types of the starting population and the possibility of cell type specific patterns with possibly some cell types with specific mutational patterns being reprogrammed more efficiently etc.

Response: We are very pleased to provide this additional discussion section as follows:

Page 15. What mechanisms could explain the changing heteroplasmy levels we have observed during re-programming? Rapid shifts in heteroplasmy could be due to a genetic bottleneck effect. This has been extensively studied in the female germ line where the bottleneck could be due to a reduction in mtDNA content within the cell¹, packaging of mtDNA into 'segregating units' which reduced the effective population size of mtDNA molecules², the focal replication of mtDNA in different regions of the cell³, and related mechanisms involving the fission and fusion of mitochondria⁴. A bottleneck effect could also occur at the cellular level due to the preferential replication or loss of cells containing specific mtDNA variants during reprogramming. This is supported by recent evidence in mice, where mtDNA variants have been shown to influence reprogramming efficiency from fibroblasts to iPSCs⁵. The selection could occur at several levels⁶. At the cellular level, difference in cellular proliferation or cell death could influence heteroplasmy levels in the whole population of cells⁷. Selection could also occur at the mitochondrial level, through increased biogenesis or selective destruction by autophagy or mitophagy⁸. Some mtDNA variants could also influence mtDNA replication itself, particularly those in the non-coding D-loop. Given emerging evidence that cell culture conditions such as the oxygen level of the addition of antioxidants can influence heteroplasmy levels⁹, including during iPSCs generation⁵, it is possible that the precise conditions used to generate the iPSCs influenced the heteroplasmy dynamics that we observed. However, the HipSci resource provides an opportunity to study all of these mechanisms now that the mtDNA variation has been characterised in detail.

Reviewer: Essential explanation of certain methods is lacking and makes this study difficult to replicate. In general, the authors tend to describe what they did, not how the analysis was performed. Which R-packages were used? Relevant information such as how iPSC cells were derived (sendai, mrna, lentiviral...), culture conditions (low oxygen/normoxia), how DNA was isolated, library prep, sequencing etc is lacking. The authors now refer to the HiPSCi website, but this is not a preferred method to describe these methods, because it doesn't encourage readers to take along this relevant information for accurate interpretation. Moreover, after publication, the paper may outlive the website.

Response: In the revised manuscript we have included details of the R-packages used in this analysis and how to access them in the Methods. We have also cited the two papers describing how the iPSC lines were generated in the HipSci project. These are open access on-line publications.

Page 16. Details of the generation of the iPSC lines including fibroblast isolation, iPSC derivation, iPSC culture, iPSC line selection, molecular assays and cell culture for maintenance and differentiation in scRNA sequencing are available at <http://www.hipsci.org/> and Refs ^{10, 11}.

On Page 21 we have included a “Code Availability” section in the main text with a web-link.

Reviewer: All mutations detected in iPSCs, but not in fibroblasts should not be called “de novo” unless solid evidence is provided these were indeed absent in the parental population.

Response: This point is addressed above, with the new discussion section shown above. The key sentence addressing the reviewer's concerns is as follows:

Page 14. Technical limitations in the sequencing approach, such as the introduction of base-errors, mean that we are unable to absolutely exclude the possibility of any variant present in iPSCs being present in fibroblasts, and our reported de novo mutation rate must be interpreted in this context.

Reviewer: In line with this: the increase in mean HF per iPSC cell line suggest mtDNA variants have become detectable as a result of the reprogramming bottleneck.

Response: We agree with the reviewer. To emphasise this point we have added this text:

Page 7. A similar genetic bottleneck effect could explain why variants not detectable in fibroblasts become detectable in iPSCs.

Reviewer: The iPSC cell lines were not clonally expanded and should therefore not be referred to as clones throughout the manuscript.

Response: We agree, and have changed each reference to iPSC clones to iPSC lines throughout the manuscript (The exception being the last section, where we defined iPSC clones based on their single cell mtDNA profile).

Reviewer: The iPSC cell lines were sequenced at highly different passages (8-43). I expect the late passage lines two have become more clonal due to drift or selection. This may influence HF detection. Please comment and discuss in manuscript.

Response: To address the reviewer's concern we have compared the mean HF, the mean number of heteroplasmies, and the mean number of *de novo* mutations between different passage groups. There was no detectable difference in any of these parameters between

the different passages. We have included this result in the main text as follows, adding a new supplementary figure:

Page 6. There was no detectable difference in either the mean HF, the mean number of heteroplasmic variants or the mean number of de novo mutations between iPSC lines analysed at different passages ($P = 1$, pairwise Wilcoxon test) (Supplementary Figs 1c & 4).

Reviewer: In the results and also the M&M section (phrased a little differently) the authors mention that “at least one iPSC line from 83 donors (56.9%) and 2 iPSC lines from 58 donors (39.7%)” were included. This is misleading, because now it seems there were $(83*1) + (58*2) = 193$ donors. Rephrase, for example: From 25 donors one iPSC line was included and from 58 donors 2 iPS cell lines were included.

Response: We thank the reviewer for the suggestion. We’ve rephrased the text as suggested.

Page 3: 141 iPSC lines and 146 fibroblast lines were included in the analysis (see Methods), with one iPSC line from 25 donors and 2 iPSC lines from 58 donors included.

Page 17: From 25 donors one iPSC line was included and from 58 donors 2 iPSC lines were included.

Reviewer: Replace “reference sequence (rCRS)” with “revised Cambridge Reference Sequence (rCRS)”.

Response: We have corrected this as suggested.

Page 4: we identified high-quality mtDNA variants relative to the revised Cambridge Reference Sequence (rCRS), including variants in mixed proportions in the bulk sequencing (heteroplasmic variants).

Page 19: Mutational spectra were derived from the revised Cambridge Reference Sequence (rCRS) and alternative alleles at each variant site.

Reviewer: Mention the absolute numbers of heteroplasmic mtDNA variants found in fibroblasts and in iPSCs in the results section, just prior to the mean number of variants.

Response: To address the reviewer’s concern we have added a new supplementary figure (Supplementary Fig. 3b) to show the absolute number of heteroplasmic variants detected in fibroblast cell and iPSC lines.

Reviewer: How are the mutation frequencies (mutation rates per base pair per genome) calculated? Were the number of mitochondria per cell taken into account (I think they should), and if so, how were these determined?

Response: The mutation rate was calculated as the number of *de novo* mutation detected / (the number of iPSC lines * 16569 (bp)) and expressed as a rate per base pair. As explained above, we did not take into account the number of mitochondria.

Reviewer: Could the authors justify the use of Stouffer’s method? The P-value from this method doesn’t seem to reflect the few more C>A, C>G, and T>A variants in iPS cells. Since the trinucleotide context is not taken into account in their conclusions, the authors could also use a Pearson’s chi square test on mutation spectra instead.

Response: Stouffer’s method is a standard approach to combine independent P-values when

looking at an overall distribution. Calculating individual P-values for each proportion in the distribution leads to a multiple-significance testing problem, thus leading to false positive results. Stouffers method minimises the type 1 error. We also used Fishers exact test, which is more reliable than the chi squared test for sub-comparisons, as follows:

Page 10: By contrast, the iPSCs had a distinct trinucleotide mutational signature (iPSCs vs fibroblasts $P = 3.66 \times 10^{-42}$, Stouffer's method for combining Fisher P values), with more C>A, C>G, T>A variants ($P = 9.47 \times 10^{-7}$, 95% CI 0.036 – 0.304, $P = 3.00 \times 10^{-30}$, 95% CI 0.020 – 0.088, $P = 0.03$, 95% CI 0.193 – 0.968, Fisher Exact Test) and less T>C variants ($P = 4.07 \times 10^{-26}$, 95% CI 2.81 - 4.73, Fisher Exact Test) than donor fibroblasts (Fig. 3e).

Reviewer: Why were indels not analysed, also because these are relevant for the mismatch repair signature that seems to be present in the mitochondria of iPS cells?

Response: We did not include indels because the calling algorithms are not as reliable as single nucleotide variant calling algorithms.

Reviewer: Page 18: replace “36,044 cell lines” with “36,044 cells”

Response: We have made this correction

Reviewer: Page 31 figure 4b is unclear. Enlarge and add absolute values

Response: We have enlarged Fig. 4c and added the absolute values.

Reviewer: FigS1A increase bar sizes of histograms to fill gaps

Response: We have adjusted Supplementary Fig. 1a as proposed.

Reviewer: Page 23 ref 4 is incomplete

Response: We have updated this reference

Reviewer #2

Reviewer: 1) At the end of the second paragraph of the introduction the authors explain that mtDNA mutations influence the transcriptomic profile of the cells and quoted two papers that focused only in tumours. However this influence is also apparent in healthy cells in an organism as it was described in mice (PMID: 32832682; PMID: 27383793).

Response: We thank the reviewer, and have included these additional important citations.

Reviewer: 2) At the end of the first paragraph in Results section the authors state: “We then determined frequency distribution of major population-specific mtDNA haplogroups (macro-haplogroups), confirming that the vast majority (98%) of donors were representative of the UK population (Figure S1B). As a further QC step, we only included heteroplasmic variants with bulk heteroplasmy allele fractions (HFs) between 2% and 98%”

Mitochondrial haplogroups U and K are very often named as UK haplogroup. However, UK does not mean United Kingdom when referring to mtDNA haplogroups. I suggest that to avoid confusion UK population here would better be expressed as United Kingdom population.

Response: We agree with the reviewer, and have changed this throughout the manuscript.

Reviewer: 3) Figure 2 B, C, E and F represent linear regression models. The statistical signification and the quality of the model of correlation estimated should be included also in the figure. In this respect it seems clear that the linear regression model does not explain well the relation between the compared variable in those graphs even when the correlation is significant. I believe that this need to be discussed.

Response: We agree with the reviewer and have included the statistical significances and the correlation estimates on the figures as requested. As we mentioned in the text (Page 4), the age correlation in fibroblasts was largely driven by the signal in the D-loop.

Reviewer: 4) Equally, the segmentation in D-loop, rRNA, tRNA and Protein encoded (synonymous vs non-synonymous) is natural but given the amount of data and because the structure of the rRNAs, tRNAs and the 13 proteins encoded is known additional relevant information can be retrieved. In the rRNA and tRNA observed mutations: is there any difference between double vs single strand structural areas? Withing the proteins coding genes: Is there any difference in the protein according to the complex where they belong (CI; CIII; CIV; CV)?

Response: To address the reviewers concern we have compared the frequency of variants in the different respiratory complex genes. The shared variants were less likely to be present in Complex III compared to the remainder ($P = 0.007$, 95% CI 0.048 - 0.760, Fisher's exact test) (**Fig. R1**). The mean number of heteroplasmic variants per fibroblast line in Complex I genes increased with the age of the donor ($P = 0.011$, coefficient estimate = 0.037, s.d = 0.014, linear regression model). There was no detectable correlation between the mean HF of any respiratory complex genes in fibroblast line with age ($P > 0.05$, linear regression model) (**Fig. R2a**). There was also no detectable correlation of the mean number of heteroplasmic variants per iPSC line of any respiratory complex genes with age, however we detected a weak correlation of the mean HF per iPSC line of the Complex I variants with the age of the donor ($P = 0.04$, coefficient estimate = 0.039, s.d = 0.0019, linear regression model) (**Fig. R2b**).

We were not planning to include these new analyses in the manuscript because we suspect that low numbers of variants in each category explain the lack of clear correlations. We could, however, add the figures below to the supplementary information if the reviewer or the editor requests this.

Fig. R1. Frequency of heteroplasmic variants in different respiratory chain complex genes. Shared, lost and *de novo* variants are shown separately.

Fig. R2. a Correlation between the mean number of heteroplasmy variants per fibroblast cell (left) and iPSC line (right) in mitochondrial respiratory complex genes and the donor's age at donation. **b** Correlation between the average heteroplasmic fraction per fibroblast cell (left) and iPSC line (right) in each mitochondrial respiratory complex genes and the donor's age at donation. Shaded regions show mean \pm standard deviation

Reviewer: 5) Regarding the mechanism of generation of de novo mutations upon iPS reprogramming the authors concluded that "To determine whether some of the de novo mutations were actually present in the fibroblasts, but below our detection threshold ($HF < 2\%$), we re-analysed the fibroblast data without a lower cut-off filter for HFs. This showed that a minority (14.8%; 40 / 270) of the previously classified de novo variants were actually present in the original fibroblast line at $HF < 2\%$. Factoring this into account, the de novo mutation rate was 8.62×10^{-5} per base pair per genome per reprogramming, which is a conservative estimate based on mutations with a $HF \geq 2\%$ in the iPSCs and a $HF > 0$ in their matched fibroblast cell lines."

The reduction of the cut-off below 2% allow to detect that 15% of the mutations considered de novo should be reclassified as pre-existing. This is a clear warning regarding the origin of all de novo classified mutations since the experimental approach is not capable to guaranty that all pre-existing mutations could be detected regardless their initial frequency since under a given threshold it would be not possible to determine errors vs true pre-existing mutations. Which is the threshold cut-off that you estimate that can be unequivocally identify a pre-existing mutation? Which is the probability that a pre-existing mutation would remain undetectable? Those are interesting estimations since your analysis raised the question of the mechanism of de novo generation mutations and if the mtDNA replication fidelity is relaxed in a programmed fashion at any stage of pluripotency. Would you comment on these issues?

Response: To address this point we have added the following paragraph to the discussion (in response to Reviewer 1).

Page 14. It is important to note that, as with any sequencing experiment, there are technical limitations with our approach. For example, we are unable to completely exclude rare variants in the fibroblast lines that fell well below the detection threshold of 2% used in the bulk WGS, either because they are at low heteroplasmy levels in many cells, or because a very small number of cells carried homoplasmic variants. Technical limitations in the sequencing approach, such as the introduction of base-errors, mean that we are unable to absolutely exclude the possibility of any variant present in iPSCs being present in fibroblasts, and our reported de novo mutation rate must be interpreted in this context. A more sensitive detection technique might reveal ultra-rare mutations in fibroblasts that are present in iPSCs at much higher levels, lowering the de novo mutation rate. However, it is also likely that a more sensitive technique would detect additional mutations in iPSCs not present in the fibroblasts, counterbalancing this effect. At this point, we can only speculate what the implications of a more sensitive detection method might be. However, this does not change our main conclusion that the heteroplasmy levels change during cell reprogramming.

To reassure the reviewer, we have also repeated our analysis using different mutation detection thresholds in iPSCs.

This shows that, as expected, reducing the detection threshold increases the measured mutation rate, but the difference is well within one order of magnitude. Thus, although the definition of the threshold does influence the results, this is minor, and does not change the overall conclusions of the manuscript.

We have included this new figure in the manuscript with the following text:

*Page 7. Finally, to determine whether our initial detection threshold had a major impact on the measured de novo mutation rate, we decreased the detection threshold in iPSCs by 0.5% increments. This showed the anticipated increased mutation rate, but only by ~4-fold, indicating that our approach is reliable to within one order or magnitude (**Supplementary Fig. 3c**).*

Reviewer: 6) Regarding the mechanism of mutation selection and generation upon reprogramming the manuscript does not explain the culture conditions of the cells nor that of the reprogramming that may be of critical relevance for the interpretation of the observations. Thus, it is known that in conventional culture conditions cells are maintain in 20% oxygen while in the body the oxygen concentration does not goes over 5%. It is also described that low oxygen concentration improve significantly genetic stability (PMID: 22139129) and the efficiency of iPS formation from fibroblast when performed at 4% Oxigen vs 20% Oxigen

(PMID: 31588014) the distribution of heteroplasmic mtDNA between fibroblast and iPS cells if performed and that culture in the presence of antioxidants prevents the shift in heteroplasmy between fibroblast and derived iPS cells (PMID: 31588014). Given the warning message derived of this work it would be of interest commenting on this issues.

Response: We agree, and have directly cited the relevant publications describing the laboratory methods in detail. These papers are open access and on line. We have also pointed out in the revised discussion that the culture conditions may be relevant for the selection/heteroplasmy shifts, citing the additional references listed by the reviewer as follows:

Page 15. Given emerging evidence that cell culture conditions such as the oxygen level of the addition of antioxidants can influence heteroplasmy levels⁹, including during iPSCs generation⁵, it is possible that the precise conditions used to generate the iPSCs influenced the heteroplasmy dynamics that we observed. However, the HipSci resource provides an opportunity to study all of these mechanisms now that the mtDNA variation has been characterised in detail.

Reviewer: 7) The discussion between “genetic bottleneck” and “cellular bottleneck” is quite interesting and is, as suggested by the reviewers related with the efficiency of reprogramming. In this respect there are strong evidence that demonstrate that the mtDNA complement determine efficiency of reprogramming from mouse fibroblast to iPS cells (PMID: 31588014).

Response: We thank the reviewer for pointing this out, and have added this point to the text:

Page 15. A bottleneck effect could also occur at the cellular level due to the preferential replication or loss of cells containing specific mtDNA variants during reprogramming. This is supported by recent evidence in mice, where mtDNA variants have been shown to influence reprogramming efficiency from fibroblasts to iPSCs⁵.

Reviewer: 8) This reviewer has problem to understand the dichotomic criteria of classification of the mutations in iPSC as increasing HF or decreasing HF without considering a third category of non-sifting. In fact, I would find that a high number of them would be better classified in this category. Which is the proportion of shifting that guaranty that a significant HF shift has been produced? I strongly believe that this data should be revisited considering the three categories. Moreover, why the authors considered that the fact the behaviour of HF for shared heteroplasmic variants between iPSCs and fibroblast argue in favour of a genetic bottleneck vs a cellular bottleneck?

Response: The HFs were measured to 1 decimal place and did not see any variant with exactly the same HF in the fibroblast line and the matched iPSC line.

Reviewer: 9) The authors notice that: “The preferential loss of D-loop variants and propagation of non-synonymous and tRNA variants are the complete opposite of what is seen during germ-line transmission”. I do not understand why they manifest surprise for that since reprogramming imply inverting the direction from germ-line mtDNA transmission and it has been described that pluripotent cells modulate heteroplasmy differently than mouse embryonic fibroblast from and that reprogramming the MEFs to iPS recapitulate the pluripotent cell behaviour (PMID: 31588014).

Response: We agree with the reviewer and have referred to this point in our discussion as follows:

Page 9. This is in keeping with work in mice, where pluripotent cells modulate heteroplasmy in a different way to mouse embryonic fibroblasts (MEFs), and reprogramming MEFs to

*iPSCs recapitulates the pluripotent cell behaviour*⁵.

Reviewer #3

Reviewer: This is an important and timely result for several reasons -- the broader mystery of heterogeneity in iPSCs and the specific usage of iPSCs in mitochondrial investigations, both of which are informed by this work. I greatly enjoyed this paper, which is clearly written (notably so, given the amount of information it conveys), with clear analyses, and with data presented well in the figures (I am particularly grateful for the distributional detail with which important quantities are plotted). I have a set of minor points but only four larger-scale ones (marked with ** below, and summarised here) before recommending acceptance.

Response: We thank the reviewer for his strong endorsement of our work

Reviewer: 1. Although most methodological choices are described in admirable detail, there is one quantity with potentially high influence that is less clear -- the lower cutoff used when characterising variants as arising *de novo* vs present in the original. Some more discussion and investigation of the influence of this choice would be useful.

Response: We chose a 2% cut off for our initial analysis based on our previous published work¹², where we analysed samples sequenced twice with ~2000-fold mtDNA coverage. This showed >95% reproducibility to detect variants present at >1% heteroplasmy fraction (HF). The data in our current submission had a lower mtDNA coverage of ~1000-fold (Supplementary Fig. 2), prompting us to use a higher, more conservative detection threshold of 2%. We have explained this more clearly in the methods section as follows:

Page 17. Our initial detection threshold of 2% heteroplasmy fraction (HF) was based on our previous work¹², where we analysed samples sequenced twice with ~2000-fold depth and showed >95% reproducibility to detect a HF > 1%. Given the mean mtDNA sequencing depth of ~1000-fold in the current dataset, we set a more conservative detection threshold of 2% HF.

We have also studied the effect of varying the detection threshold in iPSCs on the measured *de novo* mutation rate.

As expected, this shows that lowering the threshold does increase the measured, but well within one order of magnitude. This does not alter our overall conclusions, and throughout the discussion we have emphasised the importance of interpreting our findings in the context of our stated bioinformatic parameters.

We have included this new figure in the manuscript with the following text:

Page 7. Finally, to determine whether our initial detection threshold had a major impact on the measured de novo mutation rate, we decreased the detection threshold in iPSCs by 0.5% increments. This showed the anticipated increased mutation rate, but only by ~4-fold, indicating that our approach is reliable to within one order or magnitude (Supplementary Fig. 3c).

Reviewer: 2. The argument about a cellular bottleneck would seem to warrant more support. I believe the story but at the moment the copy number measurements don't necessarily support the argument in the way that is discussed.

Response: We agree, and have expanded our discussion of the potential mechanisms in the discussion citing literature in mice which supports the proposed mechanism. Our work cataloguing the mtDNA variants in the HipSci resource will enable the community to explore the underlying mechanisms in the available cell lines. The new discussion is as follows:

Page 15. What mechanisms could explain the changing heteroplasmy levels we have observed during re-programming? Rapid shifts in heteroplasmy could be due to a genetic bottleneck effect. This has been extensively studied in the female germ line where the bottleneck could be due to a reduction in mtDNA content within the cell¹, packaging of mtDNA into 'segregating units' which reduced the effective population size of mtDNA molecules², the focal replication of mtDNA in different regions of the cell³, and related mechanisms involving the fission and fusion of mitochondria⁴. A bottleneck effect could also occur at the cellular level due to the preferential replication or loss of cells containing specific mtDNA variants during reprogramming. This is supported by recent evidence in mice, where mtDNA variants have been shown to influence reprogramming efficiency from fibroblasts to iPSCs⁵. The selection could occur at several levels⁶. At the cellular level, difference in

cellular proliferation or cell death could influence heteroplasmy levels in the whole population of cells⁷. Selection could also occur at the mitochondrial level, through increased biogenesis or selective destruction by autophagy or mitophagy⁸. Some mtDNA variants could also influence mtDNA replication itself, particularly those in the non-coding D-loop. Given emerging evidence that cell culture conditions such as the oxygen level or the addition of antioxidants can influence heteroplasmy levels⁹, including during iPSCs generation⁵, it is possible that the precise conditions used to generate the iPSCs influenced the heteroplasmy dynamics that we observed. However, the HipSci resource provides an opportunity to study all of these mechanisms now that the mtDNA variation has been characterised in detail.

Reviewer: 3. I couldn't find a link to the code for the authors' pipeline to reproduce the analysis and figures. This should be provided, or a statement should be made describing why it can't be.

Response: On Page 21 we have included a "Code Availability" section in the main text with a web-link.

Reviewer: 4. There are several instances where parametric tests (often t-tests) are used when their assumptions (often normality) are clearly not met by the data. These should be replaced with non-parametric alternatives (in most cases it is clear from the data that this will not greatly affect the results). Also I request rewording several instances of "no correlation", "no effect" etc to "no detectable correlation", "no detectable effect" and similar, reflecting the fact that absence cannot be claimed based on a high p-value.

Response: We have revised the statistical analysis using non-parametric tests as requested below. For the second point, we have changed the text as requested, referring to 'no detectable difference' and 'no detectable correlation' throughout the manuscript.

Reviewer: l39 -- "similar experimental outcomes" -- it's not clear whether this means that (i) a similar amount of phenotypic difference is observed between experiments, or that (ii) phenotypes are similar between experiments (i.e. less difference). can this be clarified?

Response: We mean that the results from independent experiments are similar. We have made this more explicit as follows:

Page 2. However, independently differentiating the same cell line several times reproduces similar cell phenotypes

Reviewer: l63 -- "extreme mtDNA diversity" -- a summary comparison would be useful here. what magnitude of what measure of diversity, compared to what we see physiologically in humans?

Response: The comparison here is with the current knowledge base which assumes the mtDNA sequence is similar in cells from the same organism, hence there is no direct comparator.

Reviewer: l79 -- an odd test. are you using $p = 0.05$ to support an absence of a difference? it would be best to give the actual depth numbers here so the reader can compare for themselves.

Response: We have provided the read depths and carried out a non-parametric test.

Page 4. There was no detectable difference in the depth of mtDNA sequencing between the fibroblasts and their derived iPSCs (median depth in fibroblasts 777x, median depth in iPSCs 887x, $P = 0.907$, paired Wilcoxon test)(Supplementary Fig. 2j).

Reviewer: l81 -- expanding and citing rCRS would help for non-experts

Response: We have made this correction and cited cCRS in the main text.

Page 4. we identified high-quality mtDNA variants relative to the revised Cambridge Reference Sequence (rCRS)¹³, including variants in mixed proportions in the bulk sequencing (heteroplasmic variants).

Reviewer: l87 -- Fig S1B does not by itself support this claim that the sample is representative of the UK. I believe it from looking at the haplogroup makeup, but please provide a population distribution or other evidence for comparison, and explain where the 98% figure comes from.

Response: We thank the reviewer for highlighting this. The revised text reads as follows, including a significance test against published haplogroup data for the UK population:

*Page 4. We then determined frequency distribution of major population-specific mtDNA haplogroups (macro-haplogroups) (**Supplementary Fig. 1b**), and compared this to published data¹⁴ ($P < 2.2 \times 10^{-16}$, $R^2 = 0.99$, Pearson's correlation test), confirming that the vast majority of donors were representative of the United Kingdom population. 143 of the 146 donors (98%) belonged to one of the macro-haplogroups H, I, J, K, T, U, V or W.*

Reviewer: l100 -- please replace "a greater number" with "a great mean number" as -- if I am understanding -- your regression model looks at mean number, not number

Response: We have made the suggested correction.

Reviewer: l102 -- the test here seems different from the claim being made. from Fig 2C it indeed looks like D-loop heteroplasmy increases with age, but so does heteroplasmy in other regions, and you haven't explicitly tested whether the D-loop increase is steeper than others? of course "largely" could mean anything but the current phrasing implies that the D-loop is dominant. can you either demonstrate this, or rephrase e.g. "due in part to an increase..."

Response: We have rephrased this section in the way proposed.

Reviewer: l112 -- quick typo Figures -> Figure

Response: We have corrected this typo.

Reviewer: l117 -- "Ultimately" and "initially" contrast a bit confusingly here. could the "ultimately" be omitted?

Response: We have deleted the word 'Ultimately'

Reviewer: l130 -- quick typo "analysis *of* high-depth"

Response: We have corrected this typo.

Reviewer: l131 (and l75) -- did I miss why these 83 were chosen in particular? any risk of bias here?

Response: We only had iPSC WGS data from 83 of the 146 fibroblast cell lines.

Reviewer: I134-137 -- "no correlation", "this was not significant" -- please replace with "no detectable correlation" and "our observations did not provide statistical support for this trend". It looks to me as if there actually might be a trend underlying the fractions in Fig S3A; and in general of course we can't claim that there is no correlation just because we have a high p-value!

Response: We have edited the text exactly as suggested.

Reviewer: I140 -- I have no doubt that this is a true result but a t-test is not appropriate here as the distributions are clearly not normal. Please replace with a non-parametric test for robustness.

Response: We have performed a non-parametric test. This did not change the interpretation of the comparison.

Page 6: Overall, the iPSC lines contained less heteroplasmic variants than their fibroblast parental lines ($P < 2.2 \times 10^{-16}$, paired Wilcoxon test)

Reviewer: I150 -- is this pointing to the correct SI figure? I couldn't see mutation counts in S1C? also as above please replace I148 "no correlation" with "no detectable correlation"

Response: We thank the reviewer for pointing this out. We have added an additional figure showing this result (Supplementary Fig. 4c).

Reviewer: I173 -- quick (possible?) typo -- fibroblasts -> fibroblast

Response: We have corrected this typo.

Reviewer: ** I185 -- this approach would seem to warrant more description. what was the new cut-off you used? if you used a yet lower one, would you see that still more variants were actually present in the original lines? what is the tradeoff in terms of false positives or other issues -- ie why not use the lowest conceivable cutoff? this would seem to have a major influence on what is classified as de novo and what is transmitted.

Response: We partly answered this point above. We have also added the following text to the discussion to ensure the reader is reminded to interpret our results in the context of our analysis parameters.

Page 14. It is important to note that, as with any sequencing experiment, there are technical limitations with our approach. For example, we are unable to completely exclude rare variants in the fibroblast lines that fell well below the detection threshold of 2% used in the bulk WGS, either because they are at low heteroplasmy levels in many cells, or because a very small number of cells carried homoplasmic variants. Technical limitations in the sequencing approach, such as the introduction of base-errors, mean that we are unable to absolutely exclude the possibility of any variant present in iPSCs being present in fibroblasts, and our reported de novo mutation rate must be interpreted in this context. A more sensitive detection technique might reveal ultra-rare mutations in fibroblasts that are present in iPSCs at much higher levels, lowering the de novo mutation rate. However, it is also likely that a more sensitive technique would detect additional mutations in iPSCs not present in the fibroblasts, counterbalancing this effect. At this point, we can only speculate what the implications of a more sensitive detection method might be. However, this does not change our main conclusion that the heteroplasmy levels change during cell reprogramming.

Reviewer: ** I198-200 -- I'm not really convinced by this. the fact that you see higher copy

numbers "after" than "before" doesn't really support the presence of a reprogramming bottleneck -- it just reflects how things are in fibroblasts. there could be an argument that the (random) expansion of mtDNA from fibroblast to iPSC contributes genetic variance (see e.g. reamplification models in <https://www.frontiersin.org/articles/10.3389/fcell.2019.00294/full>) -- but the differences in copy number look too small for this to be the case. I'm an advocate of random turnover contributing genetic variation, which could apply here, as could a (not mutually exclusive) variant of the Wai-Teoli-Shoubridge picture <https://www.nature.com/articles/ng.258?foxtrotcallback=true> where only a subset of mtDNAs are "allowed" to replicate. those processes can generate genetic variability (a genetic bottleneck) without necessitating copy number reduction (a physical bottleneck). It will be clear from my comments (and my signature) that I am not a stem cell expert! Does the absence of physical sub-cloning mean that these iPSCs are not generated / maintained through repeated cell divisions? If so, could you say this explicitly, to help readers like me? If not -- random partitioning of mtDNA at cell divisions during iPSC generation / maintenance is another potential source of genetic variance.

Response: We agree with the reviewer. We have no direct evidence to support the mechanism in this paper. To address the uncertainty, we have expanded our discussion of potential mechanisms in the main discussion section as explained in our earlier response.

Reviewer: l200 -- a t-test doesn't seem appropriate here given how non-normal the distributions are. please replace with a non-parametric alternative.

Response: We have carried out this analysis using a non-parametric test.

Page 7: the average amount of mtDNA per cell was greater in iPSCs than in the matched fibroblast lines (P = 0.08, paired Wilcoxon test).

Reviewer: l201 -- quick typo -- precise -> precisely

Response: We have corrected this typo.

Reviewer: l221 -- a t-test doesn't seem appropriate here given how non-normal the distributions are. please replace with a non-parametric alternative.

Response: We have carried out this analysis using a non-parametric test.

Page 8: the mean HF was significantly higher in iPSC lines than the fibroblast lines (individual fibroblast-iPSC pairs P = 3.32 x 10⁻¹⁰, paired Wilcoxon test)

Reviewer: l224 -- not quite sure what "completely out with" means here

Response: We have re-phrased this as follows:

Page 8, ..lying outside the overall distribution.

Reviewer: l227-233 -- When you say "HS decrease", I think this means that HS were more likely to be negative, meaning a decrease of heteroplasmic fraction? At the moment it reads like the shifts were lower in magnitude -- ie a smaller change. Same for "increase".

Response: We agree, and have rephrased this as follows:

Page 8. HS were more likely to be negative than positive on reprogramming

Reviewer: l293 -- quick typo -- citation styled inconsistently with other citations

Response: We have corrected this typo.

Reviewer: l316 -- how can your definition of homoplasmy be HF > 95% if you are detecting and analysing variants between your lower cutoff of 2% and 5%?

Response: We chose a 95% cut-off to define homoplasmy in the scRNAseq data for the following reason:

The sequence data generated by low coverage scRNAseq is of lower quality than high depth WGS¹⁵ (Supplementary Fig. 2a, c). We tested this by analysing scRNAseq data in samples harbouring known homoplasmic variants detected by high depth WGS. This showed that scRNAseq variants present at >95% HF were actually homoplasmic variants by WGS (**Fig. R3** grey + purple bars). We therefore used a cut off of 95% for the scRNAseq to define homoplasmic variants in the single cells.

Fig. R3. Frequency of cells from the same iPSC line carrying the same homoplasmic variant. Each bar represents each homoplasmic variants detected by WGS. Grey bars show the proportion of the single cells from the same iPSC line that carried the same variant with HF > 98% in scRNAseq data. Grey + purple bars show the proportion of the single cells from the same iPSC line carried the same variant with HF > 95% in scRNAseq data. This showed that scRNAseq variants present at >95% HF were actually homoplasmic variants by WGS.

Reviewer: l325 -- "HF=0" -- should this be "detected HF=0" or "HF < [cutoff]"?

Response: We have re-phrased this in the text as "detected HF=0".

Reviewer: l341 -- quick typo -- "estimated *the* proportion"

Response: We have corrected this typo.

Reviewer: l352 -- quick typo -- "at a" -> "at"

Response: We have corrected this typo.

Reviewer: l371 -- please replace "no difference" with "no detectable difference"

Response: We have made this correction.

Reviewer: l374 -- quick typo -- "lineage profile*s"

Response: We have corrected this typo.

Reviewer: l384 -- I like this analysis but it must be noted that the amount of variance explained by all of these features is pretty low (the residuals are correspondingly pretty high). Can you give a ballpark for this in the main text?

Response: The average amount of variance explained by cell developmental stage was 8.8%, the experimental batch was 3.7% and cell line of origin was 1.6%. This is shown in Supplementary Fig 6d.

Reviewer: l392 -- FDR < 0.1 seems quite permissive given that you've used $p = 0.05$ to imply absence of an effect elsewhere. is there a reason for this choice?

Response: To reassure the reviewer we have re-performed the pathway and ontology analysis using FDR < 0.05, and still see the same signals, e.g respiratory electron transport, ATP biosynthesis and telomere maintenance and organisation (see updated Table S5).

In our original analysis, we did ontology analysis and found that the differentially expressed genes played a key role in mitochondrial ATP synthesis, oxidative phosphorylation, epidermal cell differentiation, and telomere maintenance and organisation. We have now included a pathway analysis and further show that the differentially expressed genes are involved in respiratory electron transport, ATP biosynthesis, gluconeogenesis, oxidative phosphorylation, differentiation and hypoxia signalling.

Reviewer: l484 -- quick typo -- "Figures" -> "Figure"

Response: We have corrected this typo.

Reviewer: l490 -- please replace "no different" with "not detectably different"

Response: We have made this correction.

Reviewer: l491 -- please replace "no correlation" with "no detectable correlation"

Response: We have made this correction.

Reviewer: l517 -- there is an official citation for R, which the initiative prefers you to use

Response: We have corrected the citation for R in the main text.

Reviewer: l618 -- please cite the UMAP algorithm (I think it's an arxiv preprint)

Response: We have cited the UMAP algorithm in the main text.

Reviewer: ** l648 -- I couldn't find a link to code to reproduce the analysis and figures of the paper. Please provide this (or state why it is not available).

Response: We have added a link to code used in this study.

References cited in the response to reviewers

(note that the numbers differ from the main text)

1. Cree, L.M. *et al.* A reduction of mitochondrial DNA molecules during embryogenesis explains the rapid segregation of genotypes. *Nat Genet* **40**, 249-254 (2008).
2. Cao, L. *et al.* New evidence confirms that the mitochondrial bottleneck is generated without reduction of mitochondrial DNA content in early primordial germ cells of mice. *PLoS Genet* **5**, e1000756 (2009).
3. Wai, T., Teoli, D. & Shoubridge, E.A. The mitochondrial DNA genetic bottleneck results from replication of a subpopulation of genomes. *Nat Genet* **40**, 1484-1488 (2008).
4. Burr, S.P., Pezet, M. & Chinnery, P.F. Mitochondrial DNA Heteroplasmy and Purifying Selection in the Mammalian Female Germ Line. *Dev Growth Differ* **60**, 21-32 (2018).
5. Latorre-Pellicer, A. *et al.* Regulation of Mother-to-Offspring Transmission of mtDNA Heteroplasmy. *Cell Metab* **30**, 1120-1130 e1125 (2019).
6. Stewart, J.B. & Chinnery, P.F. The dynamics of mitochondrial DNA heteroplasmy: implications for human health and disease. *Nat Rev Genet* **16**, 530-542 (2015).
7. Stamp, C. *et al.* Age-associated mitochondrial complex I deficiency is linked to increased stem cell proliferation rates in the mouse colon. *Aging Cell* **20**, e13321 (2021).
8. Diot, A. *et al.* Modulating mitochondrial quality in disease transmission: towards enabling mitochondrial DNA disease carriers to have healthy children. *Biochem Soc Trans* **44**, 1091-1100 (2016).
9. Estrada, J.C. *et al.* Culture of human mesenchymal stem cells at low oxygen tension improves growth and genetic stability by activating glycolysis. *Cell Death Differ* **19**, 743-755 (2012).
10. Kilpinen, H. *et al.* Common genetic variation drives molecular heterogeneity in human iPSCs. *Nature* **546**, 370-375 (2017).
11. Cuomo, A.S.E. *et al.* Single-cell RNA-sequencing of differentiating iPS cells reveals dynamic genetic effects on gene expression. *Nat Commun* **11**, 810 (2020).
12. Wei, W., Gomez-Duran, A., Hudson, G. & Chinnery, P.F. Background sequence characteristics influence the occurrence and severity of disease-causing mtDNA mutations. *PLoS Genet* **13**, e1007126 (2017).
13. Andrews, R.M. *et al.* Reanalysis and revision of the Cambridge reference sequence for human mitochondrial DNA [letter]. *Nat Genet* **23**, 147 (1999).
14. Royrvik, E.C., Burgstaller, J.P. & Johnston, I.G. mtDNA diversity in human populations highlights the merit of haplotype matching in gene therapies. *Mol Hum Reprod* **22**, 809-817 (2016).
15. Ludwig, L.S. *et al.* Lineage Tracing in Humans Enabled by Mitochondrial Mutations and Single-Cell Genomics. *Cell* **176**, 1325-1339 e1322 (2019).

REVIEWER COMMENTS

Reviewer #1 (Remarks to the Author):

While the authors have adequately addressed most of my comments, they have not convinced me that their main conclusion, namely that cell reprogramming “shapes” the mutational landscape, is supported by their findings. The authors acknowledge that their experimental approach has technical limitations and that they cannot exclude the possibility that any of the mutations that they refer to as “de novo” could also be present in the parental population below the detection threshold. Even at a sequencing depth of 2000x, most mitochondrial DNA molecules that were present in the cells used for the original DNA isolation will be missed, meaning that probably most variants are not covered at all, even not below the 2% HF. Moreover, the observed variants could also reflect the adaptation of cells to culture conditions and not be caused by the reprogramming process itself. In my previous report, I proposed to the authors to use two consecutive clonal steps that would help to discriminate between these options and would provide more insight into the processes that really shape the mitochondrial DNA landscape. As the authors indicate, this may in theory still not yield a conclusive answer, because heteroplasmy levels may be too dynamic (which would show from the results). However, as the authors demonstrate themselves, heteroplasmy levels in iPSC cells are very stable and this may also be true for the parental fibroblasts. If so, the experiment I proposed would provide solid evidence for reprogramming shaping the mutational landscape. Of course, as long as this experiment is not performed, we will never know. I realize that the authors may have a bioinformatic background and may not be in the position to perform this critical experiment. Therefore, I propose an alternative informative (but less conclusive) approach that may better suit their background. If the mutational landscape in the iPSCs is caused by the reprogramming process, then similar results can be expected if the iPSCs were to be derived from a different source of somatic cells. To my knowledge, the hIPSCI consortium has also derived iPSCs from B-lymphocytes and from fibroblasts from the same individuals, and for all three cell types sequencing data is available (doi: <https://doi.org/10.1101/2021.02.04.429731>). The resources to perform these analyses should therefore be readily available to the authors. Otherwise, other publicly available data of human iPSCs and parental cells (not fibroblasts) could be used. If the authors fail to validate their findings using such an approach, I am of the opinion that they should remove any allusions that the mitochondrial DNA variants are induced by the reprogramming processes. This specifically means replacing the misleading term “de novo”, for example with iPSC-associated mitochondrial variants. On the other hand, if the authors recapitulate their findings with a different tissue of origin, this would certainly strengthen their study, even though this approach has the same cellular bottleneck inherent to reprogramming (with a more heterogeneous parental population and a more homogeneous iPSC cell population, which will affect variant calling). The drawback of this bottleneck can probably only be circumvented by performing two consecutive clonal steps.

In response to my first comment, the authors state in their rebuttal that their approach satisfied reviewers 2 and 3. In my opinion this is not a valid argument to rebut a critical comment, because that would imply only comments raised by all reviewers need to be addressed.

The authors refer to 2 papers for details on the culture conditions of the cells. I think this is not sufficient. Essential steps/conditions that affect mutation accumulation (to provide an obvious example: oxygen concentration) need explanation so readers do not have to go look for the information elsewhere.

Page 7 line 216: change “...genetic bottleneck...” to “...cellular bottleneck...”

Page 10 lines 310-311: Mutational signatures are defined by the balance in DNA damage and repair.

Since the mitochondrial genome is not bound to histones, probably more exposed to oxidative stress, and not subjected to the same extent to DNA repair as the nuclear genome, I think that mutational patterns in mitochondrial genomes cannot readily be compared to mutational signatures extracted from nuclear genomes. I am therefore of the opinion that the observed patterns cannot be linked to disordered MMR.

Page 11 lines 324-337 I found this part a little confusing because the numbers in the text don't match with those in figure 4b, while such a graphic presentation would help.

Page 12 line 355: change "... were present at least..." to "... were present at at least..."

Page 12 line 361: change "...present more..." to change "...present in more..."

The authors calculate the number of mutations per base pair per genome without adjusting for the number of mitochondria per cell. I think mutation rates should be adjusted to the number of mitochondria per cell, because the more mitochondria there are in a cell, the higher a chance that one of these acquires a mutation.

The authors conclude that iPSCs have more mitochondria than the parental fibroblasts (Figure 1m). How accurate can the number of mitochondria (Figure 1m) be determined from WGS data, also considering the huge amount of variation in coverage of MT genome as opposed to the nuclear genome?

Related to that: if iPSCs have more mitochondria, how does that influence variant calling?

Figure 1a may be improved by showing that the scRNAseq was performed on differentiating iPSCs

I noticed that the hIPSCI consortium is no longer part of the authors list. I am curious to learn why this change was made.

Ewart Kuijk

Reviewer #2 (Remarks to the Author):

I believe that the authors have satisfactorily answer all my comments.

Reviewer #3 (Remarks to the Author):

The authors have addressed my concerns and I particularly appreciate the transparent discussion of these points in the Discussion section (and the Github repo).

Just one final point from me. The authors' heteroplasmic shift (HS) is motivated by the need to normalise the "final" heteroplasmy with respect to the "initial" heteroplasmy in the founder line. However, unless I am misunderstanding, their HS still has some dependence on the initial heteroplasmy. To see this first consider the population fraction $p(t)$ of an allele under selection in an infinite haploid population in continuous time t :

$$p(t) = 1 / (1 + (1-p(0))/p(0) \exp(-st))$$

where s is a selection coefficient. The HS definition, if I am understanding, is

$$HS = \log_2 (\text{final HF} / \text{initial HF})$$

which, setting final HF = $p(t)$, initial HF = $p(0)$, is

$$\log_2 (\exp(-st) / (1 + p(0)*(exp(-st) - 1)))$$

i.e. still dependent on initial HF $p(0)$. An inverse logit-like function can be used to remove this dependence (see e.g. Eqns 15-16 here [https://www.cell.com/ajhg/fulltext/S0002-9297\(16\)30397-4](https://www.cell.com/ajhg/fulltext/S0002-9297(16)30397-4)):

$$HS' = \log((p(t)*(p(0)-1)) / (p(0)*(p(t)-1))) \\ = st$$

hence just reflecting the action of selection over time, independent of initial HF.

In other words, the authors' HS definition serves to reflect a fold change in HF, but this fold change is itself influenced by the initial HF. A note to this effect would help interpretation of the HS statistic.

Iain Johnston

Response to Reviewers of Version R1

Reviewer #1

Reviewer: While the authors have adequately addressed most of my comments, they have not convinced me that their main conclusion, namely that cell reprogramming “shapes” the mutational landscape, is supported by their findings.

Response: We appreciate the reviewer’s concern which relates to the interpretation of the phrase ‘shapes the mutational landscape’. We deliberately chose this term to capture all of our findings, and particularly the change mutational heteroplasmy levels which occurs during reprogramming. In other words, that reprogramming ‘shapes the landscape of mutation heteroplasmy levels’. We also show that some mutations are lost from the fibroblasts on reprogramming, again providing evidence that the mutational landscape is shaped. The title of the manuscript does not actually refer to mutation events: ‘Cell reprogramming shapes the mitochondrial DNA landscape’. We hope the reviewer finds this acceptable.

Reviewer: The authors acknowledge that their experimental approach has technical limitations and that they cannot exclude the possibility that any of the mutations that they refer to as “de novo” could also be present in the parental population below the detection threshold. Even at a sequencing depth of 2000x, most mitochondrial DNA molecules that were present in the cells used for the original DNA isolation will be missed, meaning that probably most variants are not covered at all, even not below the 2% HF. Moreover, the observed variants could also reflect the adaptation of cells to culture conditions and not be caused by the reprogramming process itself. In my previous report, I proposed to the authors to use two consecutive clonal steps that would help to discriminate between these options and would provide more insight into the processes that really shape the mitochondrial DNA landscape. As the authors indicate, this may in theory still not yield a conclusive answer, because heteroplasmy levels may be too dynamic (which would show from the results). However, as the authors demonstrate themselves, heteroplasmy levels in iPSC cells are very stable and this may also be true for the parental fibroblasts. If so, the experiment I proposed would provide solid evidence for reprogramming shaping the mutational landscape. Of course, as long as this experiment is not performed, we will never know. I realize that the authors may have a bioinformatic background and may not be in the position to perform this critical experiment. Therefore, I propose an alternative informative (but less conclusive) approach that may better suit their background. If the mutational landscape in the iPSCs is caused by the reprogramming process, then similar results can be expected if the iPSCs were to be derived from a different source of somatic cells. To my knowledge, the hiPSCI consortium has also derived iPSCs from B-lymphocytes and from fibroblasts from the same individuals, and for all three cell types sequencing data is available (doi: <https://doi.org/10.1101/2021.02.04.429731>). The resources to perform these analyses should therefore be readily available to the authors. Otherwise, other publicly available data of human iPSCs and parental cells (not fibroblasts) could be used. If the authors fail to validate their findings using such an approach, I am of the opinion that they should remove any allusions that the mitochondrial DNA variants are induced by the reprogramming processes. This specifically means replacing the misleading term “de novo”, for example with iPSC-associated mitochondrial variants. On the other hand, if the authors recapitulate their findings with a different tissue of origin, this would certainly strengthen their study, even though this approach has the same cellular bottleneck inherent to reprogramming (with a more heterogeneous parental population and a more homogeneous iPSC cell population, which will affect variant calling). The drawback of this bottleneck can probably only be circumvented by performing two consecutive clonal steps.

Response: We recognise and respect the reviewer’s concerns, and have followed the reviewer’s recommendations to address them in three different ways:

(1) *Reproducing our findings in an independent dataset*

Dan Gaffney, who leads the HipSci consortium, is one of our co-authors. We have also been in contact with the senior author of the pre-print cited by the reviewer on BioRxiv (Serena Nik-Zainal). Both confirm that iPSCs have not been generated from both B-lymphocytes and fibroblasts from multiple individuals (the cited paper refers to one person only). We have, however, analysed another

public dataset (Ref ¹) which does include iPSCs derived from human blood. 10 iPSC lines from each one of 3 blood samples from 3 people. Applying our methods to their data, they reported 44 mtDNA single nucleotide variants (Heteroplasmic fraction $\geq 2\%$) in the 30 iPSC lines that were not present in the blood samples. This gives a mutation rate of 8.85×10^{-5} /base pair, which is almost identical to our measured rate of 8.62×10^{-5} /base pair (we are unable to perform the correction for copy number on this analysis because that information is not available). We hope this provides the reviewer with some reassurance that our findings are reproducible.

We have added the following text to the Discussion to reflect this:

Page 15. A re-analysis of published data¹ from human 30 iPSC clones derived from three blood samples yields a very similar mutation rate (8.85×10^{-5} /base pair) to our result calculated from the reprogramming of human fibroblasts (8.62×10^{-5} /base pair). This adds weight to our conclusion that reprogramming shapes the mtDNA landscape, and appears to be independent of the original cell type. However, definitive evidence will require a much more extensive comparison of iPSCs derived from different tissues from the same individual.

(2) Reproducing our findings in independent experiments from HipSci data

In the revised submission we included the results from 58 donors where two iPSC lines were independently derived from the parental fibroblast lines. 172 variants were found in the two iPSC lines that were not detectable in the fibroblasts, even when the detection threshold was dropped from 2% to zero. Importantly, 93% of these variants (160/172) were only detected in one of the iPSC lines, and not the other.

This result would not be expected if reprogramming was only pushing heteroplasmy levels up from low (undetectable) to high (detectable) levels. If reprogramming was simply driving up pre-existing heteroplasmies, then the majority of variants in the above experiments would be shared by both iPSC lines.

However, we agree with the reviewer that we cannot be absolutely certain that every one of the 160 variants is *de novo*, but it is highly likely that some of them are genuinely *de novo* because they cannot be seen in the other derived iPSC line. Thus, we argue that we have 'proven the principle' that new mutations are likely to have been introduced during reprogramming.

(3) Deleting reference to the term 'de novo'

We must emphasise that we completely understand the reviewer's point of view. We have therefore followed his advice and deleted all reference to the term '*de novo*', replacing this with 'iPSC-specific' mutations which we define based on the detection thresholds specified.

We have also added the following text to the discussion to address the reviewer's concerns:

Page 14. Technical limitations in the sequencing approach, such as the introduction of base-errors, mean that we are unable to absolutely exclude the possibility of any variant present in iPSCs being present in fibroblasts, and our reported iPSC-specific mutation rate must be interpreted in this context. For this reason, we have not referred to these variants as 'de novo', because we cannot be certain that these variants are not present in a very small number of molecules in the original fibroblast line.

Reviewer: In response to my first comment, the authors state in their rebuttal that their approach satisfied reviewers 2 and 3. In my opinion this is not a valid argument to rebut a critical comment, because that would imply only comments raised by all reviewers need to be addressed.

Response: We apologise. We were only trying to make the point that two other expert reviewers we satisfied with that point in our manuscript.

Reviewer: The authors refer to 2 papers for details on the culture conditions of the cells. I think this is not sufficient. Essential steps/conditions that affect mutation accumulation (to provide an obvious example: oxygen concentration) need explanation so readers do not have to go look for the information elsewhere.

Response: We have expanded the methods section as suggested (Page 16, an addition of 337 words).

Reviewer: Page 7 line 216: change "...genetic bottleneck..." to "...cellular bottleneck..."

Response: We have deleted 'genetic' to avoid confusion, reflecting the fact that the bottleneck could be either genetic or cellular.

Reviewer: Page 10 lines 310-311: Mutational signatures are defined by the balance in DNA damage and repair. Since the mitochondrial genome is not bound to histones, probably more exposed to oxidative stress, and not subjected to the same extent to DNA repair as the nuclear genome, I think that mutational patterns in mitochondrial genomes cannot readily be compared to mutational signatures extracted from nuclear genomes. I am therefore of the opinion that the observed patterns cannot be linked to disordered MMR.

Response: We have changed the wording as follows:

Page 10. The pattern of iPSC-specific mutations resembles the signature seen in the nuclear genome when there is disordered mismatch repair (MMR)² (Fig. 3g), although the absence of MMR within mitochondria implicates alternative mechanisms.

Reviewer: Page 11 lines 324-337 I found this part a little confusing because the numbers in the text don't match with those in figure 4b, while such a graphic presentation would help.

Response: We thank the reviewer for pointing this out. The numbers in the text were from the full dataset: 11,538 single cells, including 2711 iPSCs, 3402 mesendo, 3796 defendo and 1629 undefined. Figure 4b only shows the breakdown numbers from iPSCs, mesendo and defendo. We have clarified this in the main text.

Page 11. Fig. 4a, b shows the data of 9909 cells from the three cell stages.

Page 11. Fig. 4b shows the data of 9909 cells from the three cell stages.

Reviewer: Page 12 line 355: change "... were present at least..." to "... were present at at least..."

Response: We have made this proposed change.

Reviewer: Page 12 line 361: change "...present more..." to change "...present in more..."

Response: We have made this proposed change.

Reviewer: The authors calculate the number of mutations per base pair per genome without adjusting for the number of mitochondria per cell. I think mutation rates should be adjusted to the number of mitochondria per cell, because the more mitochondria there are in a cell, the higher a chance that one of these acquires a mutation.

Response: Presumably the reviewer is referring to the number of mtDNA molecules, and not the number of mitochondria? We have included a normalised value for the mutation rate as suggested.

Page 7. 1.87×10^{-6} per base pair per mtDNA molecule normalising for the average number of mtDNA molecules per cell.

Page 10. 6.82×10^{-8} per base pair per mtDNA molecule normalising for the average number of mtDNA molecules per cell.

Reviewer: The authors conclude that iPSCs have more mitochondria than the parental fibroblasts (Figure 1m). How accurate can the number of mitochondria (Figure 1m) be determined from WGS data, also considering the huge amount of variation in coverage of MT genome as opposed to the nuclear genome?

Response: Again, we presume that the reviewer is referring to the number of mtDNA molecules, and not the number of mitochondria. It is well established that WGS (and exome) read-depth is a reliable way to measure mtDNA levels relative to the nuclear genome (eg. PMID: 24901348, PMID: 32004343).

Reviewer: Related to that: if iPSCs have more mitochondria, how does that influence variant calling?

Response: Variant calling refers to the bioinformatic method to identify variants in sequencing data. We have no reason to expect this to differ between iPSCs and fibroblasts.

Reviewer: Figure 1a may be improved by showing that the scRNAseq was performed on differentiating iPSCs

Response: We thank the reviewer and have made this addition to Figure 1a.

Reviewer: I noticed that the hiPSCI consortium is no longer part of the authors list. I am curious to learn why this change was made.

Response: The work carried out was performed by Dr Wei, Gaffney and Chinnery in its entirety on publicly available data. Dr Gaffney represents the HipSci consortium, which we acknowledge in the manuscript. Based on this, the consortium itself does not justify authorship of this manuscript.

Reviewer #3:

Reviewer: The authors have addressed my concerns and I particularly appreciate the transparent discussion of these points in the Discussion section (and the Github repo).

Just one final point from me. The authors' heteroplasmic shift (HS) is motivated by the need to normalise the "final" heteroplasmy with respect to the "initial" heteroplasmy in the founder line. However, unless I am misunderstanding, their HS still has some dependence on the initial heteroplasmy. To see this first consider the population fraction $p(t)$ of an allele under selection in an infinite haploid population in continuous time t :

$$p(t) = 1 / (1 + (1-p(0))/p(0) \exp(-st))$$

where s is a selection coefficient. The HS definition, if I am understanding, is

$$HS = \log_2 (\text{final HF} / \text{initial HF})$$

which, setting final HF = $p(t)$, initial HF = $p(0)$, is

$$\log_2 (\exp(-st) / (1 + p(0)*(exp(-st) - 1)))$$

i.e. still dependent on initial HF $p(0)$. An inverse logit-like function can be used to remove this dependence (see e.g. Eqns 15-16 here [https://www.cell.com/ajhg/fulltext/S0002-9297\(16\)30397-4](https://www.cell.com/ajhg/fulltext/S0002-9297(16)30397-4)):

$$HS' = \log((p(t)*(p(0)-1)) / (p(0)*(p(t)-1))) \\ = st$$

hence just reflecting the action of selection over time, independent of initial HF.

In other words, the authors' HS definition serves to reflect a fold change in HF, but this fold change is itself influenced by the initial HF. A note to this effect would help interpretation of the HS statistic.

Response: We are pleased to add this additional clarification to the text as follows:

Page 8. Note that this correction reflects the fold-change in HF, but the size of this change is influenced by the initial HF.

References

1. Kang, E. *et al.* Age-Related Accumulation of Somatic Mitochondrial DNA Mutations in Adult-Derived Human iPSCs. *Cell Stem Cell* **18**, 625-36 (2016).
2. Alexandrov, L.B. *et al.* The repertoire of mutational signatures in human cancer. *Nature* **578**, 94-101 (2020).